# Continuous PDE Dynamics Forecasting with Implicit Neural Representations

**Yuan Yin**[*1]  **Matthieu Kirchmeyer**[*1,2]  **Jean-Yves Franceschi**[*2]
**Alain Rakotomamonjy**[2]  **Patrick Gallinari**[1,2]
[1]Sorbonne Université, CNRS, ISIR, F-75005 Paris, France  [2]Criteo AI Lab, Paris

## Abstract

Effective data-driven PDE forecasting methods often rely on fixed spatial and / or temporal discretizations. This raises limitations in real-world applications like weather prediction where flexible extrapolation at arbitrary spatiotemporal locations is required. We address this problem by introducing a new data-driven approach, DINo, that models a PDE's flow with continuous-time dynamics of spatially continuous functions. This is achieved by embedding spatial observations independently of their discretization via Implicit Neural Representations in a small latent space temporally driven by a learned ODE. This separate and flexible treatment of time and space makes DINo the first data-driven model to combine the following advantages. It extrapolates at arbitrary spatial and temporal locations; it can learn from sparse irregular grids or manifolds; at test time, it generalizes to new grids or resolutions. DINo outperforms alternative neural PDE forecasters in a variety of challenging generalization scenarios on representative PDE systems.

## 1 Introduction

Modeling the dynamics and predicting the temporal evolution of physical phenomena is paramount in many fields, e.g. climate modeling, biology, fluid mechanics and energy (Willard et al., 2022). Classical solutions rely on a well-established physical paradigm: the evolution is described by differential equations derived from physical first principles, and then solved using numerical analysis tools, e.g. finite elements, finite volumes or spectral methods (Olver, 2014). The availability of large amounts of data from observations or simulations has motivated data-driven approaches to this problem (Brunton & Kutz, 2022), leading to a rapid development of the field with deep learning methods. The main motivations for this research track include developing surrogate or reduced order models that can approximate high-fidelity full order models at reduced computational costs (Kochkov et al., 2021), complementing classical solvers, e.g. to account for additional components of the dynamics (Yin et al., 2021), or improving low fidelity models (De Avila Belbute-Peres et al., 2020).

Most of these attempts rely on workhorses of deep learning like CNNs (Ayed et al., 2020) or GNNs (Li et al., 2020; Pfaff et al., 2021; Brandstetter et al., 2022). They all require prior space discretization either on regular or irregular grids, such that they only capture the dynamics on the train grid and cannot generalize outside it. Neural operators, a recent trend, learn mappings between function spaces (Li et al., 2021b; Lu et al., 2021) and thus alleviate some limitations of prior discretization approaches. Yet, they still rely on fixed grid discretization for training and inference: e.g., regular grids for Li et al. (2021b) or a free-form but predetermined grid for Lu et al. (2021). Hence, the number and / or location of the sensors has to be fixed across train and test which is restrictive in many situations (Prasthofer et al., 2022). Mesh-agnostic approaches for solving canonical PDEs (Partial Differential Equations) are another trend (Raissi et al., 2019; Sirignano & Spiliopoulos, 2018). In contrast to physics-agnostic grid-based approaches, they aim at solving a known PDE as usual solvers do, and cannot cope with unknown dynamics. This idea was concurrently developed for computer graphics, e.g. for learning 3D shapes (Sitzmann et al., 2020; Mildenhall et al., 2020; Tancik et al., 2020) and coined as Implicit Neural Representations (INRs). When used as solvers, these methods can only tackle a single initial value problem and are not designed for long-term forecasting outside the training horizon.

---

[*]Equal contribution

Table 1: Comparison of data-driven approaches to spatiotemporal PDE forecasting.

| Model | Reference | 1. PDE-agnostic prediction on new initial conditions | 2. Train / test space grid independence | 3. Evaluation at unobserved spatial locations | 4. Free-form spatial domain (manifold, irregular mesh) | 5. Time continuous | 6. Time extrapolation |
|---|---|:---:|:---:|:---:|:---:|:---:|:---:|
| Discrete { NODE | Chen et al. (2018) | ✓ | ✗ | ✗ | ✗ | ✓ | ✓ |
| MP-PDE | Brandstetter et al. (2022) | ✓ | ✗ | ✗ | ✓ | ✗ | ✓ |
| Operator { MNO | Li et al. (2021a) | ✓ | ✓ | ✗ | ✗ | ✗ | ✓ |
| DeepONet | Lu et al. (2021) | ✓ | ✗ | ✓ | ✓ | ✓ | ✗ |
| INRs { PINNs | Raissi et al. (2019) | ✗ | ✓ | ✓ | ✓ | ✓ | ✗ |
| DINo | Ours | ✓ | ✓ | ✓ | ✓ | ✓ | ✓ |

Because of these limitations, none of the above approaches can handle situations encountered in many practical applications such as: different geometries, e.g. phenomena lying on a Euclidean plane or an Earth-like sphere; variable sampling, e.g. irregular observation grids that may evolve at train and test time as in adaptive meshing (Berger & Oliger, 1984); scarce training data, e.g. when observations are only available at a few spatiotemporal locations; multi-scale phenomena, e.g. in large scale-dynamics systems as climate modeling, where integrating intertwined subgrid scales a.k.a. the closure problem is ubiquitous (Zanna & Bolton, 2021). These considerations motivate the development of new machine learning models that improve existing approaches on several of these aspects.

In our work, we aim at forecasting PDE-based spatiotemporal physical processes with a versatile model tackling the aforementioned limitations. We adopt an agnostic approach, i.e. not assuming any prior knowledge on the physics. We introduce DINo (Dynamics-aware Implicit Neural representations), a model operating continuously in space and time, with the following contributions.

**Continuous flow learning.** DINo aims at learning the PDE's flow to forecast its solutions, in a continuous manner so that it can be trained on any spatial and temporal discretization and applied to another. To this end, DINo embeds spatial observations into a small latent space via INRs; then it models continuous-time evolution by a learned latent Ordinary Differential Equation (ODE).
**Space-time separation.** To efficiently encode different sequences, we propose a novel INR parameterization, amplitude modulation, implementing a space-time separation of variables. This simplifies the learned dynamics, reduces the number of parameters and greatly improves performance.
**Spatiotemporal versatility.** DINo combines the benefits of prior models; cf. Table 1. It tackles new sequences via its amplitude modulation. Sequential modeling with an ODE makes it extrapolate to unseen times within or beyond the train horizon. Thanks to INRs' spatial flexibility, it generalizes to new grids or resolutions, predicts at arbitrary positions and handles sparse irregular grids or manifolds.
**Empirical validation.** We demonstrate DINo's versatility and state-of-the-art performance versus prior neural PDE forecasters, representative of grid, operator and INR-based methods, via thorough experiments on challenging multi-dimensional PDEs in various spatiotemporal generalization settings.

## 2 PROBLEM DESCRIPTION

**Problem setting.** We aim at modeling, via a data-driven approach, the temporal evolution of a continuous fully-observed deterministic spatiotemporal phenomenon. It is described by trajectories $v \colon \mathbb{R} \to \mathcal{V}$ in a set $\Gamma$; we use $v_t \triangleq v(t) \in \mathcal{V}$. We focus on Initial Value Problems, where only $v_t$ at any time $t$ is required to infer $v_{t'}$ for $t' > t$. Hence, trajectories share the same dynamics but differ by their initial condition $v_0 \in \mathcal{V}$. $\mathbb{R}$ is the temporal domain and $\mathcal{V}$ is the functional space of the form $\Omega \to \mathbb{R}^n$, where $\Omega \subset \mathbb{R}^p$ is a compact spatial domain and $n$ the number of observed values. In other words, $v_t$ is a spatial function of $x \in \Omega$, with vectorial output $v_t(x) \in \mathbb{R}^n$; cf. examples of Section 5.1. To this end, we consider the setting illustrated in Figure 1. We observe a finite training set of trajectories $\mathcal{D}$ with a free-form spatial observation grid $\mathcal{X}_{\mathrm{tr}} \subset \Omega$ and on discrete times $t \in \mathcal{T} \subset [0, T]$. At test time, we are only given a new initial condition $v_0$, with observed values $v_0|_{\mathcal{X}_{\mathrm{ts}}}$ on a new observation grid $\mathcal{X}_{\mathrm{ts}}$, potentially different from $\mathcal{X}_{\mathrm{tr}}$. Inference is performed on both train and test trajectories given only the initial condition, on a new free-form grid $\mathcal{X}' \subset \Omega$ and times $t \in \mathcal{T}' \subset [0, T']$. Inference grid $\mathcal{X}'$ comprises observed positions (respectively $\mathcal{X}_{\mathrm{tr}}$ and $\mathcal{X}_{\mathrm{ts}}$ for train and test trajectories) and unobserved positions corresponding to spatial interpolation. Note that the inference temporal horizon is larger than the train one: $T < T'$. For simplicity, *In-s* refers to data in $\mathcal{X}'$ on the observation grid ($\mathcal{X}_{\mathrm{tr}}$ for *train* / $\mathcal{X}_{\mathrm{ts}}$ for *test*), *Out-s* to data in $\mathcal{X}'$ outside the observation grid; *In-t* refers to times within the train horizon $\mathcal{T} \subset [0, T]$, and *Out-t* to times in $\mathcal{T}' \setminus \mathcal{T} \subset (T, T']$, beyond $T$, up to inference horizon $T'$.

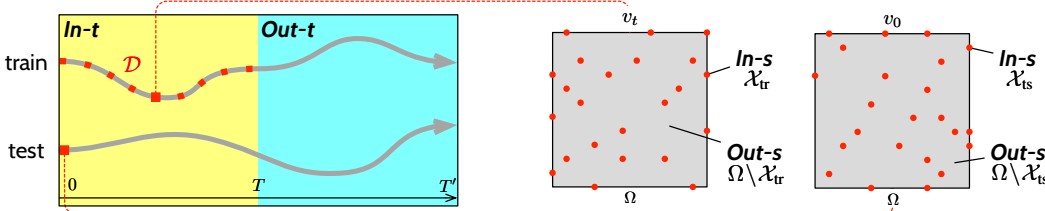

Figure 1: (Left) We represent time contexts. The *train* trajectory consists of training snapshots (■), observed in a train interval $[0, T]$ denoted *In-t*. The line (—) in continuation is a forecasting of this trajectory beyond *In-t*, in $(T, T')$ denoted *Out-t*. The line below (—, *test*) is a forecasting from a new initial condition $v_0$ (■) on *In-t* and *Out-t*. (Middle and right) We illustrate spatial contexts. (Middle) Dots (•) correspond to the train observation grid $\mathcal{X}_{tr}$, denoted *In-s*. *Out-s* denotes the complementary domain $\Omega \setminus \mathcal{X}_{tr}$. (Right) New test observation grid $\mathcal{X}_{ts}$, used as an initial point for forecasting (left).

**Evaluation scenarios.** The desired properties in Section 1 call for spatiotemporally continuous forecasting models. We select six criteria that our approach should meet; cf. column titles of Table 1. First, the model should be robust to the change of initial condition $v_0$, i.e. generalize to *test* trajectories (col. 1). Second, it should extrapolate beyond the train conditions: in space, on a test observation grid that differs from the train one, i.e. $\mathcal{X}' = \mathcal{X}_{ts} \neq \mathcal{X}_{tr}$ (*In-s*) (col. 2), and outside the observed train and test grid, i.e. on $\mathcal{X}' \setminus \mathcal{X}_{ts}, \mathcal{X}' \setminus \mathcal{X}_{tr}$ (*Out-s*, col. 3); in time, between train snapshots (col. 5) and beyond the observed train horizon $T$ (*Out-t*, col. 6). Finally, it should adapt to free-form spatial domains, i.e. to various geometries (e.g. manifolds) or irregular grids (col. 4). See also Figure 1.

**Objective.** To satisfy these requirements, we learn the flow $\Phi$ of the physical system:

$$\Phi \colon (\mathcal{V} \times \mathbb{R}) \to \mathcal{V}, \qquad (v_t, \tau) \mapsto \Phi_\tau(v_t) = v_{t+\tau} \quad \forall v \in \Gamma, t \in \mathbb{R}. \tag{1}$$

Learning the flow is a common strategy in sequential models to better generalize beyond the train time horizon. Yet, so far, it has always been learned with discretized models, which poses generalization issues violating our requirements. We describe these issues in Section 3.

## 3 RELATED WORK

We review current data-driven approaches for PDE modeling and the representative methods listed in Table 1. We express the forecasting rule using the notations in Eq. (1): $t$ is an arbitrary time; $\tau$ is an arbitrary time interval; $\delta t$ is a fixed, predetermined time interval (as a model hyperparameter).

**Sequential discretized models.** Most sequential dynamics models are learned on a fixed observed grid $\mathcal{X}_{tr}$ and use discretized models, e.g. CNN or GNN to process the observations. CNNs require observations on a regular grid but can be extended to irregular grids through interpolation (Chae et al., 2021). GNNs are more flexible as they handle irregular grids, at an additional memory and computational cost. Yet, prediction on new grids $\mathcal{X}' \neq \mathcal{X}_{tr}$ fails experimentally for both CNNs and GNNs, as these discretized models are biased towards the training grid $\mathcal{X}_{tr}$, as we later show in Section 5. We distinguish two types of temporal models which both extrapolate beyond the train horizon due to their sequential nature. • Autoregressive models $v_t|_{\mathcal{X}} \mapsto v_{t+\delta t}|_{\mathcal{X}}$ (Long et al., 2018; de Bézenac et al., 2018; Pfaff et al., 2021; Brandstetter et al., 2022) predict the sequence from $t$ only at fixed time increments $\delta t$ and not in between. • Time-continuous extensions using numerical solvers $(v_t|_{\mathcal{X}}, \tau) \mapsto v_{t+\tau}|_{\mathcal{X}}$ (Yin et al., 2021; Iakovlev et al., 2021) solve this limitation as they provide a prediction at arbitrary times, thus remove dependency on the time discretization.

**Operator learning.** Recently, operator-based models aim at finding a parameterized mapping between functions. They define in theory space-continuous models. First, neural operators (Kovachki et al., 2021) attempt to replace standard convolution with continuous alternatives. Fourier Neural Operator (FNO, Li et al., 2021b) applies convolution in the spectral domain via Fast Fourier Transformation (FFT). Graph Neural Operator (GNO, Li et al., 2020) performs convolution on a local interaction grid described by a graph. Second, DeepONet (Lu et al., 2021) uses a coordinate-based neural network to output a prediction at arbitrary time and space locations given a function observed on a fixed grid. Three types of temporal models were used for operators with some limitations. • The standard approach, $v_0 \mapsto v_t$, models the output at a given time $t \in [0, T]$ within the train horizon (Li et al., 2020). • A sequential extension, $v_t \mapsto v_{t+\delta t}$, was proposed in Li et al. (2021a). • Finally, a time-continuous

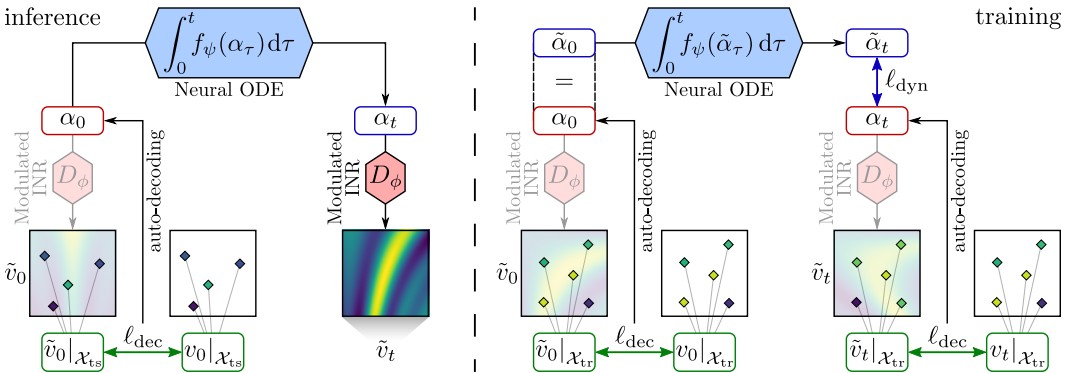

Figure 2: Proposed DINo model. Inference (left): given a new initial condition observed on a grid $\mathcal{X}_{ts}$, $v_0|_{\mathcal{X}_{ts}}$, forecasting amounts at decoding $\alpha_t$ to $\tilde{v}_t$, by unrolling $\alpha_0$ with a time-continuous ODE dynamics model $f_\psi$. Train (right): given an observation grid $\mathcal{X}_{tr}$ and a space-continuous decoder $D_\phi$, $\alpha_t$ is learned by auto-decoding s.t. $D_\phi(\alpha_t)|_{\mathcal{X}_{tr}} = v_t|_{\mathcal{X}_{tr}}$; its evolution is then modeled with $f_\psi$.

version $v_0 \mapsto (t \in [0, T] \mapsto v_t)$ in DeepONet propose a solution at arbitrary time and space locations. The first and third approaches are not designed to generalize beyond the train horizon, i.e. when $t > T$ as they are not sequential. The second solves this limitation but is only able to predict solutions from $t$ at fixed time increments of $\delta t$ and not in-between. Furthermore, all existing approaches make restrictive assumptions on the space discretization. They lack flexibility when encoding spatial observations: FNO is limited to uniform Cartesian observation grids due to FFT, and while one concurrent follow-up alleviates this issue (Li et al., 2022), it still cannot perform predictions on unobserved spatial locations; GNO does not adapt well to changing observation grids as for the GNN-based models in the previous paragraph; DeepONet is limited to input observations on fixed observation locations. The latter are chosen at random spatial positions but should remain fixed throughout training and testing.

**Spatiotemporal INRs.** Another class of models relies on coordinate-based neural networks, called Implicit Neural Representations (INRs, Sitzmann et al., 2020; Fathony et al., 2021; Tancik et al., 2020). These space-continuous models share a similar objective as operators, despite constituting a separate research field. INRs for spatiotemporal data take time as an input along spatial coordinates. Physics-informed neural networks (PINNs, Raissi et al., 2019) use this formulation to solve PDEs, yet are limited to a single known differential equation and a set of initial and boundary conditions. Fresca et al. (2020) and Chen et al. (2023) combine INRs with reduced order models. Extensions for multi-sequence learning, e.g. for video generation (Yu et al., 2022; Skorokhodov et al., 2022) or compression (Chen et al., 2021), learn a latent conditioning variable from an initial condition $v_0$, i.e. take the form $v_0 \mapsto (t \in [0, T] \mapsto v_t)$. Interestingly, these models can predict at an arbitrary time $t$ in the train horizon without unrolling a sequential model up to $t$. Yet, as they only learn mappings from an initial condition $v_0$ to a function of time $v_t$ in the train domain, they fail to predict beyond train conditions, as we show in Section 5. DINo is a new instance of spatiotemporal INR which solves this limitation via a time-continuous dynamics model of the underlying flow, $(v_t, \tau) \mapsto v_{t+\tau}$.

# 4 MODEL

We present DINo, the first space / time-continuous model that tackles all prediction tasks of Section 2, without the above limitations. We specify DINo's inference procedure (Section 4.1), illustrated in Figure 2 (left), then introduce each of its components (Section 4.2) and how they are trained (Section 4.3, Figure 2 (right)). Finally, we detail our implementation based on amplitude modulation, a novel INR parameterization for spatiotemporal data which performs separation of variables (Section 4.4).

## 4.1 INFERENCE MODEL

As explained in Section 2, we aim at estimating the flow $\Phi$ in Eq. (1), so that our model can be trained on an observed grid $\mathcal{X}_{tr}$ and perform inference given a new one $\mathcal{X}_{ts}$, both possibly irregular. To this end, we leverage a space- and time-continuous formulation, independent of a given data discretization. At inference, DINo starts from a single initial condition $v_0 \in \mathcal{V}$ and uses a flow to

forecast its dynamics. DINo first embeds spatial observations from $v_0$ into a latent vector $\alpha_0$ of small dimension $d_\alpha$ via an encoder of spatial functions $E_\varphi \colon \mathcal{V} \to \mathbb{R}^{d_\alpha}$ (ENC). Then, it unrolls a latent time-continuous dynamics model $f_\psi \colon \mathbb{R}^{d_\alpha} \to \mathbb{R}^{d_\alpha}$ given this initial condition (DYN). Finally, it decodes latent vectors via a decoder $D_\phi \colon \mathbb{R}^{d_\alpha} \to \mathcal{V}$ into a spatial function (DEC). At any time $t$, $D_\phi$ takes as input $\alpha_t$ and outputs a function $\tilde{v}_t \colon \Omega \to \mathbb{R}^n$. This results in the following model, illustrated in Figure 2 (left) and whose components are detailed in Section 4.2:

$$\text{(ENC) } \alpha_0 = E_\varphi(v_0), \qquad \text{(DYN) } \frac{\mathrm{d}\alpha_t}{\mathrm{d}t} = f_\psi(\alpha_t), \qquad \text{(DEC) } \forall t, \tilde{v}_t = D_\phi(\alpha_t). \qquad (2)$$

## 4.2 COMPONENTS

**Encoder:** $\alpha_t = E_\varphi(v_t)$. The encoder computes a latent vector $\alpha_t$ given observation $v_t$ at any time $t$. It is used in two different contexts, respectively for train and test. At train time, given an observed trajectory $v_\mathcal{T} = \{v_t\}_{t \in \mathcal{T}}$, it will encode any $v_t$ into $\alpha_t$ (see Section 4.3). At inference time, only $v_0$ is available, and then only $\alpha_0$ is computed to be used as initial value for the dynamics. Given the decoder $D_\phi$, $\alpha_t$ is a solution to the inverse problem $D_\phi(\alpha_t) = v_t$. We solve this inverse problem with auto-decoding (Park et al., 2019). Denoting $\ell_{\mathrm{dec}}(\phi, \alpha_t; v_t) = \|D_\phi(\alpha_t) - v_t\|_2^2$ the decoding loss where $\|\cdot\|_2$ is the euclidean norm of a function and $K$ the number of update steps, auto-decoding defines $E_\varphi$ as:

$$E_\varphi(v_t) = \alpha_t^K, \quad \text{where} \quad \alpha_t^0 = \alpha_t; \ \forall k > 0, \alpha_t^{k+1} = \alpha_t^k - \eta \nabla_{\alpha_t} \ell_{\mathrm{dec}}(\phi, \alpha_t^k; v_t) \quad \text{and} \quad \varphi = \phi. \quad (3)$$

In practice, we observe a discretization $(\mathcal{X}_{\mathrm{tr}}, \mathcal{X}_{\mathrm{ts}})$ and accordingly approximate the norm in $\ell_{\mathrm{dec}}$ as in Eq. (6). Compared to auto-encoding, auto-decoding underfits less (Kim et al., 2019) and is more flexible: without requiring specialized encoder architecture, it handles free-formed (irregular or on a manifold) observation grids as long as the decoder shares the same property.

**Decoder:** $\tilde{v}_t = D_\phi(\alpha_t)$. We define a flexible decoder using a coordinate-based INR network with parameters conditioned on $\alpha_t$. An INR $I_\theta \colon \Omega \to \mathbb{R}^n$ is a space-continuous model parameterized by $\theta \in \mathbb{R}^{d_\theta}$ defined on domain $\Omega$. It approximates functions independently of the observation grid, e.g. it handles irregular grids and changing observation positions unlike FNO and DeepONet. Thus, it constitutes a flexible alternative to operators suitable to auto-decoding. To implement the conditioning of the INR's parameters, we use a hypernetwork (Ha et al., 2017) $h_\phi \colon \mathbb{R}^{d_\alpha} \to \mathbb{R}^{d_\theta}$, as illustrated in Figure 3. It generates high-dimensional parameters $\theta_t \in \mathbb{R}^{d_\theta}$ of the INR given the low-dimensional latent vector $\alpha_t \in \mathbb{R}^{d_\alpha}$. Hence, the decoder $D_\phi$, parameterized by $\phi$, is defined as:

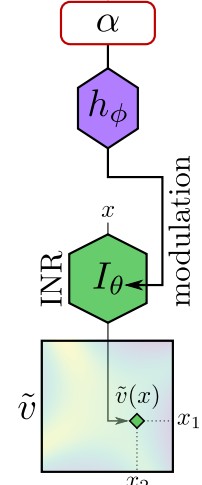

$$\forall x \in \Omega, \quad \tilde{v}_t(x) = D_\phi(\alpha_t)(x) \triangleq I_{h_\phi(\alpha_t)}(x). \qquad (4)$$

The decoder's predictions at all spatial locations $x \in \Omega$ thus all depend on $\alpha_t$. We provide further details on the precise implementation in Section 4.4.

**Dynamics model:** $\frac{\mathrm{d}\alpha_t}{\mathrm{d}t} = f_\psi(\alpha_t)$. Finally, the dynamics model $f_\psi \colon \mathbb{R}^{d_\alpha} \to \mathbb{R}^{d_\alpha}$ defines a flow via an ODE in the latent space. The initial condition can be defined at any time $t$ by encoding with $E_\varphi$ the corresponding input function $v_t$.

Figure 3: Decoding via INR Eq. (4).

**Overall flow.** Combined altogether, our components define the following flow in the input space that can approximate the data flow $\Phi$ in Eq. (1):

$$\forall (t, \tau), \qquad (v_t, \tau) \mapsto D_\phi\big(E_\varphi(v_t) + \textstyle\int_t^{t+\tau} f_\psi(\alpha_{\tau'})\,\mathrm{d}\tau'\big) \quad \text{where } \alpha_t = E_\varphi(v_t). \qquad (5)$$

To summarize, DINo defines a time-continuous latent temporal model with a space-continuous emission function $D_\phi$, combining the flexibility of space and time continuity. This is fully novel to our knowledge, as prior latent approaches are discretized (cf. Fraccaro (2018) for state-space models).

## 4.3 TRAINING

We present the training procedure, illustrated in Figure 2 (right), of the previous components. We use a two-stage optimization process, close to recent work in video prediction (Yan et al., 2021). Given the train sequences $\mathcal{D}$, we first use auto-decoding to obtain the latent vectors $\alpha_\mathcal{T} = \{\alpha_t^v\}_{t \in \mathcal{T}, v \in \mathcal{D}}$ and the decoder parameters $\phi$. We then learn the parameters of the dynamics $\psi$ by modeling the latent flow

over $\alpha_t^v, \forall v \in \mathcal{D}$. We detail this procedure in Appendix D.1, which can be formalized as a two-stage optimization problem that we solve in parallel without inducing training instability (cf. Appendix D.2):

$$\min_\psi \quad \ell_{\text{dyn}}(\psi, \alpha_\mathcal{T}) \triangleq \mathbb{E}_{v \in \mathcal{D}, t \in \mathcal{T}} \big\| \alpha_t^v - \big(\alpha_0^v + \int_0^t f_\psi(\alpha_\tau^v) \, d\tau\big) \big\|_2^2$$

$$\text{s.t. } \alpha_\mathcal{T}, \phi = \arg\min_{\alpha_\mathcal{T}, \phi} \quad \ell_{\text{dec}}(\phi, \alpha_\mathcal{T}) \triangleq \mathbb{E}_{v \in \mathcal{D}, x \in \mathcal{X}_{\text{tr}}, t \in \mathcal{T}} \big\| v_t(x) - D_\phi(\alpha_t^v)(x) \big\|_2^2. \tag{6}$$

## 4.4 DECODER IMPLEMENTATION VIA AMPLITUDE-MODULATED INRS

We now specify our implementation of decoder $D_\phi$ in Eq. (4). This includes the definition of the INR architecture $I_\theta$ and of the hypernetwork $h_\phi$. We introduce for the latter a new method called amplitude modulation, which implements a space-time separation of variables.

$I_\theta$ **as FourierNet.** We implement $I_\theta$ as a FourierNet, a state-of-the-art INR architecture, which instantiates a Multiplicative Filter Network (MFN, Fathony et al., 2021). A FourierNet relies on the recursion in Eq. (7), where $x \in \Omega$ is an input spatial location, $z^{(l)}(x)$ is the hidden feature vector at layer $l$ for $x$ and $s_{\omega^{(l)}}(x) = [\cos(\omega^{(l)}x), \sin(\omega^{(l)}x)]$ is a Fourier basis:

$$\begin{cases} z^{(0)}(x) = s_{\omega^{(0)}}(x), \qquad z^{(L)}(x) = W^{(L-1)} z^{(L-1)}(x) + b^{(L-1)}, \\ z^{(l)}(x) = \big(W^{(l-1)} z^{(l-1)}(x) + b^{(l-1)}\big) \odot s_{\omega^{(l)}}(x) \quad \text{for } l \in [\![1, L-1]\!], \end{cases} \tag{7}$$

where we fix $W^{(0)} = 0$, $b^{(0)} = 1$, $s_{\omega^{(0)}}(x) = x$ and $\odot$ is the Hadamard product. Denoting $W = [W^{(l)}]_{l=1}^{L-1}, b = [b^{(l)}]_{l=1}^{L-1}, \omega = [\omega^{(l)}]_{l=1}^{L-1}$, we fit a FourierNet to an input function $v$ observed on a grid $\mathcal{X}$ by learning $\{W, b, \omega\}$ s.t. $\forall x \in \mathcal{X}, z^{(L)}(x) = v(x)$. In practice, we observe that fixing $\omega$ uniformly sampled performs similarly as learning them, so we exclude them from training parameters.

FourierNets are interpretable, a property we leverage to separate time and space via amplitude modulation. Fathony et al. (2021) show that for some $M \gg L \in \mathbb{N}$, there exist a set of coefficients $\{c_j^{(m)}\}_{m=1}^M$ that depend individually on $\{W, b\}$ as well as a set of parameters $\{\gamma^{(m)}\}_{m=1}^M$ that depend individually on those of the filters $\omega$ s.t. the $j^{\text{th}}$ dimension of $z^{(L)}(x)$ can be expressed as:

$$z_j^{(L)}(x) = \sum_{m=1}^M c_j^{(m)} s_{\gamma^{(m)}}(x) + \text{bias}. \tag{8}$$

Eq. (8) involves a basis of spatial functions $\{s_{\gamma^{(m)}}\}_{m=1}^M$ evaluated on $x$ and the amplitudes of this basis $\{c_j^{(m)}\}_{m=1}^M$. Note that Eq. (8) can be extended to other choices of $s_{\omega^{(l)}}$ (Fathony et al., 2021).

$h$ **as amplitude modulation.** $h$ generates the INR's parameters $\theta_t$ given $\alpha_t$ to model a target input function $v_t$. We implement $h$ as elementwise shift and scale transformations (FiLM, Perez et al., 2018) of the linear layers parameters $W, b$ (excluding those of the filters $\omega$). Then, in Eq. (8), amplitudes $c_j^{(m)}$ only depend on time while the basis functions $s_{\gamma^{(m)}}$ only depend on space: this corresponds to a modeling assumption of separation of variables (Le Dret & Lucquin, 2016) in $v$. We call our technique amplitude modulation. In practice, as Dupont et al. (2022), we consider latent shift transformations (Figure 4), detailed in Eq. (9). Eq. (9) extends Eq. (7) by introducing a shift term $\mu_t^{(l-1)}$ at each layer $l$, defined as $\mu_t^{(l-1)} = W'^{(l-1)} \alpha_t$, where $W' = [W'^{(l-1)}]_{l=1}^{L-1}$ is another weight matrix:

$$z_t^{(l)}(x) = \big(W^{(l-1)} z_t^{(l-1)}(x) + b^{(l-1)} + \mu_t^{(l-1)}\big) \odot s_{\omega^{(l)}}(x). \tag{9}$$

The INR's parameters are defined as $h_\phi(\alpha_t) = \{W; b + W'\alpha_t; \omega\}$ where $\phi = \{W, b, W'\}$ are $h$'s parameters. Thus, amplitude modulation separates time and space. We show in Table 5 that it significantly improves performance, particularly time extrapolation.

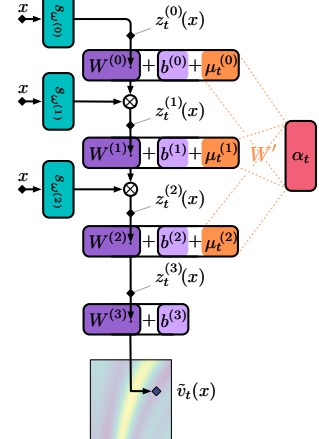

Figure 4: Amplitude modulation – Eq. (9). $z_t^{(l-1)}$ is input to the $l^{\text{th}}$ linear layer and combined with the spatial basis $s_{\omega^{(l)}}$ via Hadamard product.

## 5 EXPERIMENTS

We assess the spatiotemporal versatility of DINo, following Section 2. We introduce our experimental setting (Section 5.1), which includes a variety of challenging PDE datasets, state-of-the-art baselines and forecasting tasks. Then, we present and comment the experimental results (Section 5.2).

Table 2: **Space and time generalization.** Train and test observation grids are equal and subsampled from an uniform 64×64 grid, used for inference. We report MSE (↓) on the inference time interval $\mathcal{T}'$, divided within training horizon (*In-t*, $\mathcal{T}$) and beyond (*Out-t*, outside $\mathcal{T}$) across subsampling ratios.

| Model | Navier-Stokes | | | | Wave | | | |
| | Train | | Test | | Train | | Test | |
| | In-t | Out-t | In-t | Out-t | In-t | Out-t | In-t | Out-t |
| --- | --- | --- | --- | --- | --- | --- | --- | --- |
| *s = 5% subsampling ratio* | | | | | | | | |
| Discrete Operator { I-MP-PDE | 8.154E−3 | 8.166E−3 | 7.926E−3 | 8.225E−3 | 7.055E−4 | 7.097E−4 | 1.138E−3 | 1.116E−3 |
| DeepONet | 3.330E−3 | 7.370E−3 | 1.346E−2 | 1.408E−2 | 8.331E−4 | 9.295E−3 | 1.692E−2 | 3.256E−2 |
| INR { SIREN | 8.741E−3 | 1.767E−1 | 4.303E−2 | 2.126E−1 | 2.738E−3 | 1.818E−2 | 3.339E−2 | 6.964E−2 |
| DINo | **1.029E−3** | **1.655E−3** | **1.326E−3** | **1.813E−3** | **4.088E−5** | **4.121E−5** | **6.415E−5** | **7.392E−5** |
| *s = 25% subsampling ratio* | | | | | | | | |
| Discrete Operator { I-MP-PDE | 3.135E−4 | 7.245E−4 | 3.476E−4 | 7.658E−4 | 3.293E−5 | 1.108E−4 | 5.142E−5 | 1.545E−4 |
| DeepONet | 9.016E−4 | 5.936E−3 | 9.376E−3 | 1.328E−2 | 5.722E−4 | 1.061E−2 | 1.757E−2 | 3.221E−2 |
| INR { SIREN | 5.180E−3 | 2.175E−4 | 2.436E−4 | 3.861E−1 | 8.995E−4 | 1.292E−2 | 1.783E−2 | 5.143E−2 |
| DINo | **1.020E−4** | **4.504E−4** | **2.646E−4** | **5.951E−4** | **3.949E−6** | **4.436E−6** | **1.089E−5** | **1.174E−5** |
| *s = 100% subsampling ratio* | | | | | | | | |
| Discrete { CNODE | 2.319E−2 | 9.652E−2 | 2.305E−2 | 1.143E−1 | 2.337E−5 | 5.280E−4 | 3.057E−5 | 7.288E−4 |
| MP-PDE | 1.140E−4 | 5.500E−4 | **1.785E−4** | 5.856E−4 | **1.718E−7** | 1.993E−5 | **9.256E−7** | 4.261E−5 |
| Operator { MNO | **3.190E−5** | 8.678E−4 | 2.763E−4 | 8.946E−4 | 9.381E−6 | 4.890E−3 | 1.993E−4 | 6.128E−3 |
| DeepONet | 1.375E−3 | 6.573E−3 | 9.704E−3 | 1.244E−2 | 6.431E−4 | 1.293E−2 | 1.847E−2 | 3.317E−2 |
| SIREN | 1.066E−3 | 4.336E−1 | 3.874E−1 | 1.037E0 | 3.674E−4 | 9.956E−3 | 3.013E−2 | 7.842E−2 |
| INR { MFN | 1.651E−3 | 1.037E0 | 2.106E−1 | 1.059E0 | 1.408E−4 | 1.763E−1 | 4.735E−3 | 2.274E−1 |
| DINo (no sep.) | 3.235E−4 | 1.593E−3 | 7.850E−4 | 1.889E−3 | 2.641E−6 | 4.081E−5 | 5.977E−5 | 2.979E−4 |
| DINo | 8.339E−5 | **3.115E−4** | 2.092E−4 | **4.311E−4** | 3.309E−6 | **3.506E−6** | 9.495E−6 | **9.946E−6** |

## 5.1 EXPERIMENTAL SETTING

**Datasets.** We consider the following PDEs defined over a spatial domain $\Omega$, with further details in Appendix C. • **2D Wave equation** (*Wave*) is a second-order PDE $\frac{\partial^2 u}{\partial t^2} = c^2 \Delta u$. $u$ is the displacement w.r.t. the rest position and $c$ is the wave traveling speed. We consider its first-order form, so that $v_t = (u_t, \frac{\partial u_t}{\partial t})$ has a two-dimensional output ($n = 2$). • **2D Navier Stokes** (*Navier-Stokes*, Stokes, 1851) corresponds to an incompressible fluid dynamics $\frac{\mathrm{d}v}{\mathrm{d}t} = -u\nabla v + \nu\Delta v + f, v = \nabla \times u, \nabla u = 0$, where $u$ is the velocity field and $v$ the vorticity. $\nu$ is the viscosity and $f$ is a constant forcing term; $n = 1$. • **3D Spherical shallow water** (*Shallow-Water*, Galewsky et al., 2004): it involves the vorticity $w$, tangent to the sphere's surface, and the thickness of the fluid $h$. The input is $v_t = (w_t, h_t)$; $n = 2$.

**Baselines.** We reimplement representative models from Section 3 and Table 1 and adapt them to our multi-dimensional datasets. • **CNODE** (Ayed et al., 2020) combines a CNN and an ODE solver to handle regular grids. • **MP-PDE** (Brandstetter et al., 2022) uses a GNN to handle free-formed grids, yet is unable to predict outside the observation grid. We developed an interpolative extension, **I-MP-PDE**, to handle this limitation; it performs bicubic interpolation on the observed grid and training is done on the resulting interpolation. • **MNO** (Li et al., 2021a) is an autoregressive version of FNO (Li et al., 2021b) for regular grids; it can be evaluated on new uniform grids. • **DeepONet** (Lu et al., 2021), considered autoregressively (Wang & Perdikaris, 2021) where we remove time from the trunk net's input, can be evaluated on new spatial locations without interpolation. • **SIREN** (Sitzmann et al., 2020) and **MFN** (Fathony et al., 2021) are two INR methods which we extend to fit our setting. We consider an agnostic setting, i.e. without the knowledge of the differential equation, and perform sequence conditioning to generalize to more than a trajectory. This is achieved by learning a latent vector with auto-decoding; it is then concatenated to the spatial coordinates.

**Tasks.** We evaluate models on various forecasting tasks which combine the evaluation scenarios of Section 2. Performance is measured by the prediction Mean Squared Error (MSE) given only an initial condition. • **Space and time generalization.** We consider a uniform grid $\mathcal{X}'$ for inference. Training is performed on different observations grids $\mathcal{X}_{tr}$ subsampled from $\mathcal{X}'$ with different ratios, $s \in \{5\%, 25\%, 50\%, 100\%\}$ where $s = 100\%$ corresponds to the full inference grid, i.e. $\mathcal{X}_{tr} = \mathcal{X}'$. In this setting, we consider that all trajectories (*train* and *test*) share the same observation grid $\mathcal{X}_{tr} = \mathcal{X}_{ts}$. We evaluate MSE error on $\mathcal{X}'$ over the train time interval (*In-t*) and beyond (*Out-t*) at each subsampling ratio. • **Flexibility w.r.t. input grid.** We vary the test observation grid, i.e. $\mathcal{X}_{ts} \neq \mathcal{X}_{tr}$ and perform inference on $\mathcal{X}' = \mathcal{X}_{ts}$, i.e. on the test observation grid (*In-s*) under two settings: ▷ **Generalizing across grids:** $\mathcal{X}_{tr}, \mathcal{X}_{ts}$ are subsampled differently from the same uniform grid; $s_{tr}$ (resp. $s_{ts}$) is the train

Table 3: **Flexibility w.r.t. input grid.** Observed test / train grid differ ($\mathcal{X}_{ts} \neq \mathcal{X}_{tr}$). We report *test* MSE ($\downarrow$) for *Navier-Stokes* on $\mathcal{X}' = \mathcal{X}_{ts}$ (*In-s*). **Green** Yellow Red mean excellent, good, poor MSE.

(a) **Generalization across grids**: $\mathcal{X}_{tr}, \mathcal{X}_{ts}$ are subsampled with different ratios $s_{tr} \neq s_{ts}$ among $\{5, 50, 100\}\%$ from the same uniform $64{\times}64$ grid.

| Subsampling | Test→ | $s_{ts} = 5\%$ | | $s_{ts} = 50\%$ | | $s_{ts} = 100\%$ | |
| --- | --- | --- | --- | --- | --- | --- | --- |
| Train ↓ | | In-t | Out-t | In-t | Out-t | In-t | Out-t |
| $s_{tr} = 5\%$ | MP-PDE | 1.330E−1 | 3.852E−1 | 1.859E−1 | 6.680E−1 | 2.105E−1 | 7.120E−1 |
| | DINo | **1.494E−3** | **2.291E−3** | **1.257E−3** | **1.883E−3** | **1.287E−3** | **1.947E−3** |
| $s_{tr} = 50\%$ | MP-PDE | 4.494E−2 | 9.403E−2 | 4.793E−3 | 1.997E−2 | 6.330E−3 | 3.712E−2 |
| | DINo | **2.470E−4** | **4.697E−4** | **2.073E−4** | **4.284E−4** | **2.058E−4** | **4.361E−4** |
| $s_{tr} = 100\%$ | MP-PDE | 1.358E−1 | 3.355E−1 | 1.182E−2 | 2.664E−2 | **1.785E−4** | 5.856E−4 |
| | DINo | **2.495E−4** | **4.805E−4** | **2.109E−4** | **4.325E−4** | 2.092E−4 | **4.311E−4** |

(b) **Generalization across resolutions**: $\mathcal{X}_{ts}$ (resp. $\mathcal{X}_{tr}$) are subsampled at the same ratio $s \in \{5, 100\}\%$ from different uniform grids with resolution $r_{ts} \in \{32, 64, 256\}$ (resp. $r_{tr} = 64$).

| Test resolution → | | $r_{ts} = 32$ - $\mathcal{X}_{ts} \neq \mathcal{X}_{tr}$ | | $r_{ts} = 64$ - $\mathcal{X}_{ts} = \mathcal{X}_{tr}$ | | $r_{ts} = 256$ - $\mathcal{X}_{ts} \neq \mathcal{X}_{tr}$ | |
| --- | --- | --- | --- | --- | --- | --- | --- |
| Subsampling ↓ | | In-t | Out-t | In-t | Out-t | In-t | Out-t |
| $s = 5\%$ | MP-PDE | 3.209E−1 | 6.472E−1 | **2.465E−4** | 1.105E−3 | 2.239E−1 | 8.253E−1 |
| | DINo | **5.308E−3** | **9.544E−3** | 2.533E−4 | **8.832E−4** | **1.991E−3** | **2.942E−3** |
| $s = 100\%$ | MNO | 4.547E−3 | 9.281E−3 | **1.277E−4** | 8.525E−4 | 2.174E−3 | 4.975E−3 |
| | MP-PDE | 4.194E−2 | 9.109E−2 | 1.597E−4 | 6.483E−4 | 4.648E−2 | 1.381E−1 |
| | DINo | **2.321E−4** | **6.386E−4** | 2.320E−4 | **6.385E−4** | **2.322E−4** | **6.385E−4** |

(resp. test) subsampling ratio. ▷ **Generalizing across resolutions:** $\mathcal{X}_{tr}, \mathcal{X}_{ts}$ are subsampled with the same ratio $s$ from two uniform grids with different resolutions; the train resolution is fixed to $r_{tr} = 64$ while we vary the test resolution $r_{ts} \in \{32, 64, 256\}$. • **Data on manifold.** We consider a PDE on a sphere and combine several evaluation scenarios, as described later. • **Finer time resolution.** We consider an inference time grid $\mathcal{T}'$ with a finer resolution than the train one $\mathcal{T}$.

## 5.2 RESULTS

**Space and time generalization.** We report prediction MSE in Table 2 for varying subsampling ratios $s \in \{5\%, 25\%, 100\%\}$ on *Navier-Stokes* and *Wave*. Appendix A provides a fine-grained evaluation inside the train observation grid (*In-s*) or outside (*Out-s*) and additionally reports the results for $s = 50\%$. We visualize some predictions in Appendix B. DINo is compared to all baselines when $s = 100\%$, i.e. $\mathcal{X}' = \mathcal{X}_{tr} = \mathcal{X}_{ts}$, and otherwise it is compared only to models which handle irregular grids and prediction at arbitrary spatial locations (DeepONet, SIREN, MFN, I-MP-PDE). • **General analysis.** We observe that all models degrade when the subsampling ratio $s$ decreases. DINo performs competitively overall: it achieves the best *Out-t* performance on all subsampling settings, it outperforms all the baselines on low subsampling ratios and performs comparably to the competitive discretized (MP-PDE, CNODE) and operator (MNO) alternatives when $s = 100\%$, i.e. when observation and inference grids are equal. Note that this fully observed setting is favorable for CNODE, MP-PDE and MNO, designed to perform inference on the observation grid. This can be seen in Table 2, where DINo is slightly outperformed only for few settings. MP-PDE is significantly better only on *Wave* for *In-t*. Overall, CNNs and GNNs exhibit good performance for spatially local dynamics like *Wave*, while INRs (like DINo) and MNO are more adapted to global dynamics like *Navier-Stokes*. • **Analysis per model.** MP-PDE is the most competitive baseline across datasets as it combines a strong and flexible encoder (GNNs) to a good dynamics model; however, it cannot predict outside the observation grid (*Out-s*). To keep a strong competitor, we extend this baseline into its interpolative version I-MP-PDE on subsampled settings. I-MP-PDE is competitive for high subsampling ratios, e.g. $s \in \{50\%, 100\%\}$ but underperforms w.r.t. DINo at lower subsampling ratios due to the accumulated interpolation error. MNO is a competitive baseline on *Navier-Stokes*, performing on par with MP-PDE and DINo inside the training horizon (*In-t*); its performance on *Out-t* degrades more significantly compared to other models, especially DINo. DeepONet is more flexible than MP-PDE as it can predict at arbitrary locations. As no interpolation error is introduced, it outperforms I-MP-PDE for $s = 5\%$ on *train* data. Yet, we observe that it underperforms especially on *Out-t* w.r.t. its alternatives. Finally, we observe that SIREN and MFN fit correctly the train

horizon *In-t* on *train*, yet generalize poorly outside this horizon *Out-t* or on new initial conditions (*test*). This is in accordance with our analysis of Section 3; we highlight that this is not the case for DINO which extrapolates temporally and generalizes to new initial conditions thanks to its sequential modeling of the flow. Thus, DINO *is currently the state-of-the-art INR model for spatiotemporal data*.
• **Modulation.** We observe that modulating both amplitudes and frequencies (row "DINO (no sep.)" in Table 2) degrades performance w.r.t. DINO (row "DINO" in Table 2) that only modulates amplitudes. Amplitude modulation enables long temporal extrapolation and reduces the number of parameters. Hence, as opposed to DINO (no sep.) which is outperformed by some baselines, time-space variable separation in DINO is an essential ingredient of the model to reach state-of-the-art levels.

**Flexibility w.r.t. input grid.** We consider in Table 3 *Navier-Stokes* and compare DINO to the most competitive baselines, MP-PDE and MNO (with $s = 100\%$ subsampling ratio). • **Generalizing across grids.** In Table 3a, we consider that the test observation grid $\mathcal{X}_{ts}$ is different from the train one $\mathcal{X}_{tr}$. This occurs when sensors differ between two observed trajectories. We vary the subsampling ratio for the train observation grid $s_{tr}$ and the test one $s_{ts}$. We report *test* MSE on new grids $\mathcal{X}' = \mathcal{X}_{ts}$. We observe that DINO is very robust to changing grids between *train* and *test*, while MP-PDE's performance degrades, especially for low subsampling ratios, e.g. 5%. For reference, we report in Table 6 Appendix A (result col. 1) the performance when $\mathcal{X}' = \mathcal{X}_{tr}$, where MP-PDE is substantially better. • **Generalizing across spatial resolutions.** In Table 3b we vary the test resolution $r_{ts}$. We train at a resolution $r_{tr} = 64$ and perform inference at resolutions $r_{ts} \in \{32, 64, 256\}$. For that, we build a high-fidelity 256×256 simulation dataset and downscale it to obtain the other resolutions. We observe that DINO's performance is the stablest across resolutions in the uniform or irregular setting. MNO is also relatively stable but is only applicable to uniform grids while MP-PDE is particularly brittle, especially for a 5% ratio.

**Data on manifold.** We consider in Figure 5 *Shallow-Water* in a super-resolution setting: test resolution is twice the train one, close to weather prediction applications. We observe

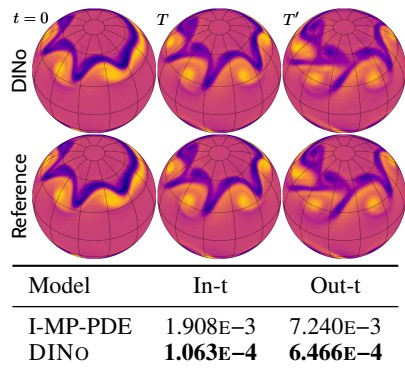

| Model | In-t | Out-t |
|---|---|---|
| I-MP-PDE | 1.908E−3 | 7.240E−3 |
| DINO | **1.063E−4** | **6.466E−4** |

Figure 5: **Data on manifold.** DINO's *Shallow-Water* superresolution *test* prediction (top) against the reference (middle), with *test* MSE (↓) (bottom).

Table 4: **Finer time resolution.** *Test* MSE (↓) under $\mathcal{T}'$ for *Navier-Stokes*.

| Model | In-t | Out-t |
|---|---|---|
| I-DINO (linear) | 3.459E−4 | 5.598E−4 |
| I-DINO (quadratic) | 2.165E−4 | 4.473E−4 |
| DINO (ODE solve) | **2.151E−4** | **4.388E−4** |

an irregular 3D Euclidean coordinate grid $\mathcal{X}_{tr} = \mathcal{X}_{ts} \subset \mathbb{R}^3$ shared for *train* and *test*. It uniformly samples Euclidean positions on the sphere, via the quasi-uniform skipped latitude-longitude grid (Weller et al., 2012). We predict the PDE on *test* trajectories with a conventional latitude-longitude inference grid $\mathcal{X}'$. At Earth scale, $\mathcal{X}_{tr}$ corresponds to a resolution of about 300 km, and $\mathcal{X}'$ to 150 km. DINO significantly outperforms I-MP-PDE, making it a viable candidate for this complex setting.

**Finer time resolution.** We consider in Table 4 a longer and ten times finer test temporal grid $\mathcal{T}'$ than the train grid $\mathcal{T}$ on *Navier-Stokes*. We observe the same spatial uniform grid across *train* and *test* and perform inference on this grid. We compare DINO that performs prediction with an ODE solver, to interpolating coarser predictions obtained at the train resolution (I-DINO). We report the corresponding *test* MSE. We observe that the ODE solver accurately extrapolates outside the train temporal grid, outperforming interpolation. This confirms that DINO benefits from its continuous-time modeling of the flow, providing consistency and stability across temporal resolutions.

## 6 CONCLUSION

We propose DINO, a novel space- and time-continuous data-driven PDE forecaster. DINO handles free-form spatiotemporal conditions encountered in many applications, where existing methods fail. DINO outperforms recent PDE forecasters on a variety of PDEs and spatiotemporal generalization settings, including evaluation on unseen sparse irregular meshes and resolutions. There are many promising future work such as scaling DINO to real-world problems, e.g. weather forecasting, or incorporating recent strategies to adapt to changing dynamics (Kirchmeyer et al., 2022).

ACKNOWLEDGEMENTS

We thank Emmanuel de Bézenac and Jérémie Donà for helpful insights and discussions on this project. We also acknowledge financial support from DL4CLIM (ANR-19-CHIA-0018-01) and DEEPNUM (ANR-21-CE23-0017-02) ANR projects. This study has been conducted using E.U. Copernicus Marine Service Information.

REPRODUCIBILITY STATEMENT

We present in Section 5.1 our experimental setting with datasets, baselines and forecasting tasks. The train and test settings are detailed in Appendix C, including more information on the chosen physical PDE systems. We describe DINo's pseudo-code in Algorithm 1 and provide implementation and hyperparameters details in Appendix D. We provide our source code at https://github.com/mkirchmeyer/DINo.

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

## A   FULL RESULTS

We provide in Table 5 a more detailed version of Table 2 for the space-time extrapolation problem where we report the performance *In-s* (on the observation grid) and *Out-s* (outside). We add $s = 50\%$.

Then, we report in Table 6 a more detailed version of Table 3a, which includes the results of $\mathcal{X}_{\text{ts}} = \mathcal{X}_{\text{tr}}$. This corresponds to our generalization across grids problem.

Table 5: **Space and time generalization.** The train and test observation grids are equal; they are subsampled with a ratio $s$ from an uniform 64×64 grid fixed here to be the inference grid $\mathcal{X}'$. We report MSE ($\downarrow$) on $\mathcal{X}'$ (on the observation grid *In-s*, outside *Out-s* or on both *Full*) and the inference time interval $\mathcal{T}'$, divided within training horizon (*In-t*, $\mathcal{T}$) and beyond (*Out-t*, outside $\mathcal{T}$) across subsampling ratios $s \in \{5\%, 25\%, 50\%, 100\%\}$. Best in **bold** and second best underlined.

| | Model | Navier-Stokes | | | | Wave | | | |
| --- | --- | --- | --- | --- | --- | --- | --- | --- | --- |
| | | Train | | Test | | Train | | Test | |
| | | In-t | Out-t | In-t | Out-t | In-t | Out-t | In-t | Out-t |
| | | | | | *s = 5% subsampling* | | | | |
| In-s | I-MP-PDE | **3.525E−5** | 1.295E−3 | 4.554E−4 | 1.414E−3 | **1.824E−6** | 8.672E−5 | 1.113E−5 | 1.987E−4 |
| | DeepONet | 4.778E−4 | 4.517E−3 | 1.060E−2 | 1.059E−2 | 2.546E−4 | 8.831E−3 | 1.501E−2 | 3.196E−2 |
| | SIREN | 5.966E−3 | 1.769E−1 | 4.082E−2 | 2.150E−1 | 1.690E−3 | 1.707E−2 | 2.951E−2 | 6.955E−2 |
| | DINo | 1.016E−4 | **6.945E−4** | **3.623E−4** | **8.306E−4** | 2.250E−6 | **5.283E−6** | **7.530E−6** | **2.146E−5** |
| Out-s | I-MP-PDE | 8.550E−3 | 8.515E−3 | 8.306E−3 | 8.571E−3 | 7.412E−4 | 7.414E−4 | 1.195E−3 | 1.163E−3 |
| | DeepONet | 3.475E−3 | 7.515E−3 | 1.361E−2 | 1.426E−2 | 8.624E−4 | 9.318E−3 | 1.702E−2 | 3.259E−2 |
| | SIREN | 8.882E−3 | 1.767E−1 | 4.314E−2 | 2.124E−1 | 2.791E−3 | 1.823E−2 | 3.359E−2 | 6.965E−2 |
| | DINo | **1.076E−3** | **1.704E−3** | **1.375E−3** | **1.863E−3** | **4.285E−5** | **4.304E−5** | **6.703E−5** | **7.659E−5** |
| Full | I-MP-PDE | 8.154E−3 | 8.166E−3 | 7.926E−3 | 8.225E−3 | 7.055E−4 | 7.097E−4 | 1.138E−3 | 1.116E−3 |
| | DeepONet | 3.330E−3 | 7.370E−3 | 1.346E−2 | 1.408E−2 | 8.331E−4 | 9.295E−3 | 1.692E−2 | 3.256E−2 |
| | SIREN | 8.741E−3 | 1.767E−1 | 4.303E−2 | 2.126E−1 | 2.738E−3 | 1.818E−2 | 3.339E−2 | 6.964E−2 |
| | DINo | **1.029E−3** | **1.655E−3** | **1.326E−3** | **1.813E−3** | **4.088E−5** | **4.121E−5** | **6.415E−5** | **7.392E−5** |
| | | | | | *s = 25% subsampling* | | | | |
| In-s | I-MP-PDE | 1.447E−4 | 5.677E−4 | **1.763E−4** | 6.147E−4 | **6.754E−7** | 8.251E−5 | **9.253E−7** | 1.227E−4 |
| | DeepONet | 7.500E−4 | 5.779E−3 | 9.227E−3 | 1.300E−2 | 5.196E−4 | 1.058E−2 | 1.743E−2 | 3.246E−2 |
| | SIREN | 4.786E−3 | 2.178E−1 | 2.461E−1 | 3.884E−1 | 8.478E−4 | 1.282E−2 | 1.733E−2 | 5.104E−2 |
| | DINo | **8.295E−5** | **4.273E−4** | 2.444E−4 | **5.735E−4** | 3.194E−6 | **3.747E−6** | 8.907E−6 | **1.029E−5** |
| Out-s | I-MP-PDE | 3.678E−4 | 7.748E−4 | 4.026E−4 | 8.143E−4 | 4.330E−5 | 1.200E−4 | 6.764E−5 | 1.648E−4 |
| | DeepONet | 9.503E−4 | 5.987E−3 | 9.423E−3 | 1.337E−2 | 5.891E−4 | 1.062E−2 | 1.762E−2 | 3.213E−2 |
| | SIREN | 5.305E−3 | 2.173E−1 | 2.428E−1 | 3.853E−1 | 9.159E−4 | 1.295E−2 | 1.798E−2 | 5.156E−2 |
| | DINo | **1.081E−4** | **4.578E−4** | **2.711E−4** | **6.021E−4** | **4.192E−6** | **4.657E−6** | **1.153E−5** | **1.220E−5** |
| Full | I-MP-PDE | 3.135E−4 | 7.245E−4 | 3.476E−4 | 7.658E−4 | 3.293E−5 | 1.108E−4 | 5.142E−5 | 1.545E−4 |
| | DeepONet | 9.016E−4 | 5.936E−3 | 9.376E−3 | 1.328E−2 | 5.722E−4 | 1.061E−2 | 1.757E−2 | 3.221E−2 |
| | SIREN | 5.180E−3 | 2.175E−1 | 2.436E−1 | 3.861E−1 | 8.995E−4 | 1.292E−2 | 1.783E−2 | 5.143E−2 |
| | DINo | **1.020E−4** | **4.504E−4** | **2.646E−4** | **5.951E−4** | **3.949E−6** | **4.436E−6** | **1.089E−5** | **1.174E−5** |
| | | | | | *s = 50% subsampling* | | | | |
| In-s | I-MP-PDE | 1.153E−4 | 5.016E−4 | **1.594E−4** | 6.043E−4 | **2.200E−7** | 3.179E−5 | **8.843E−7** | 5.854E−5 |
| | DeepONet | 6.214E−4 | 4.277E−3 | 5.699E−3 | 1.082E−2 | 7.581E−4 | 1.187E−2 | 1.649E−2 | 3.378E−2 |
| | SIREN | 4.911E−3 | 6.815E−1 | 1.607E−1 | 6.889E−1 | 5.134E−4 | 1.481E−2 | 3.086E−2 | 8.196E−2 |
| | DINo | **8.151E−5** | **2.920E−4** | 2.004E−4 | **4.283E−4** | 3.277E−6 | **3.659E−6** | 8.978E−6 | **9.572E−6** |
| Out-s | I-MP-PDE | 1.186E−4 | 5.010E−4 | **1.626E−4** | 6.132E−4 | **9.638E−7** | 3.153E−5 | **2.367E−6** | 5.574E−5 |
| | DeepONet | 6.851E−4 | 4.343E−3 | 5.740E−3 | 1.099E−2 | 7.842E−4 | 1.185E−2 | 1.679E−2 | 3.391E−2 |
| | SIREN | 5.067E−3 | 6.867E−1 | 1.599E−1 | 6.845E−1 | 5.354E−4 | 1.492E−2 | 3.113E−2 | 8.333E−2 |
| | DINo | **9.175E−5** | **3.041E−4** | 2.116E−4 | **4.409E−4** | 3.277E−6 | **3.659E−6** | 8.978E−6 | **9.572E−6** |
| Full | I-MP-PDE | 1.170E−4 | 5.013E−4 | **1.611E−4** | 6.088E−4 | **6.021E−7** | 3.166E−5 | **1.646E−6** | 5.710E−5 |
| | DeepONet | 6.541E−4 | 4.311E−3 | 5.720E−3 | 1.091E−2 | 7.715E−4 | 1.186E−2 | 1.665E−2 | 3.385E−2 |
| | SIREN | 4.995E−3 | 6.841E−1 | 1.603E−1 | 6.867E−1 | 5.246E−4 | 1.486E−2 | 3.100E−2 | 8.265E−2 |
| | DINo | **8.677E−5** | **2.982E−4** | 2.062E−4 | **4.348E−4** | 3.380E−6 | **3.751E−6** | 9.251E−6 | **9.710E−6** |
| | | | | | *s = 100% subsampling* | | | | |
| Full | CNODE | 2.319E−2 | 9.652E−2 | 2.305E−2 | 1.143E−1 | 2.337E−5 | 5.280E−4 | 3.057E−5 | 7.288E−4 |
| | MP-PDE | 1.140E−4 | 5.500E−4 | **1.785E−4** | 5.856E−4 | **1.718E−7** | 1.993E−5 | **9.256E−7** | 4.261E−5 |
| | MNO | **3.190E−5** | 8.678E−4 | 2.763E−4 | 8.946E−4 | 9.381E−6 | 4.890E−3 | 1.993E−4 | 6.128E−3 |
| | DeepONet | 1.375E−3 | 6.573E−3 | 9.704E−3 | 1.244E−2 | 6.431E−4 | 1.293E−2 | 1.847E−2 | 3.317E−2 |
| | SIREN | 1.066E−3 | 4.336E−1 | 3.874E−1 | 1.037E0 | 3.674E−4 | 9.956E−3 | 3.013E−2 | 7.842E−2 |
| | MFN | 1.651E−3 | 1.037E0 | 2.106E−1 | 1.059E0 | 1.408E−4 | 1.763E−1 | 4.735E−3 | 2.274E−1 |
| | DINo (no sep.) | 3.235E−4 | 1.593E−3 | 7.850E−4 | 1.889E−3 | 2.641E−6 | 4.081E−5 | 5.977E−5 | 2.979E−4 |
| | DINo | 8.339E−5 | **3.115E−4** | 2.092E−4 | **4.311E−4** | 3.309E−6 | **3.506E−6** | 9.495E−6 | **9.946E−6** |

Table 6: **Generalization across grids.** $\mathcal{X}_{\text{tr}}, \mathcal{X}_{\text{ts}}$ are subsampled with different ratios $s_{\text{tr}} \neq s_{\text{ts}} \in \{5, 50, 100\}\%$ from the same uniform $64 \times 64$ grid. We report *test* MSE within $\mathcal{X}_{ts}$ (*In-s*). **Best** in bold.

| Subsampling | Test → | $\mathcal{X}_{\text{ts}} = \mathcal{X}_{\text{tr}}$ | | $\mathcal{X}_{\text{ts}} \neq \mathcal{X}_{\text{tr}}$ | | | | | |
| | | $s_{\text{ts}} = s_{\text{tr}}$ | | $s_{\text{ts}} = 5\%$ | | $s_{\text{ts}} = 50\%$ | | $s_{\text{ts}} = 100\%$ | |
| Train ↓ | | In-t | Out-t | In-t | Out-t | In-t | Out-t | In-t | Out-t |
|---|---|---|---|---|---|---|---|---|---|
| $s_{\text{tr}} = 5\%$ | MP-PDE | **1.967E−4** | **6.631E−4** | 1.330E−1 | 3.852E−1 | 1.859E−1 | 6.680E−1 | 2.105E−1 | 7.120E−1 |
| | DINo | 3.623E−4 | 8.306E−4 | **1.494E−3** | **2.291E−3** | **1.257E−3** | **1.883E−3** | **1.287E−3** | **1.947E−3** |
| $s_{\text{tr}} = 50\%$ | MP-PDE | **1.346E−4** | 5.110E−4 | 4.494E−2 | 9.403E−2 | 4.793E−3 | 1.997E−2 | 6.330E−3 | 3.712E−2 |
| | DINo | 2.004E−4 | **4.283E−4** | **2.470E−4** | **4.697E−4** | **2.073E−4** | **4.284E−4** | **2.058E−4** | **4.361E−4** |
| $s_{\text{tr}} = 100\%$ | MP-PDE | **1.785E−4** | 5.856E−4 | 1.358E−1 | 3.355E−1 | 1.182E−2 | 2.664E−2 | **1.785E−4** | 5.856E−4 |
| | DINo | 2.092E−4 | **4.311E−4** | **2.495E−4** | **4.805E−4** | **2.109E−4** | **4.325E−4** | 2.092E−4 | **4.311E−4** |

# B  PREDICTION

We display the test prediction of DINo (Figure 6) and I-MP-PDE (Figure 7) for various subsampling levels when $\mathcal{X} = \mathcal{X}_{\text{tr}} = \mathcal{X}_{\text{ts}}$. Predictions are performed on a $64 \times 64$ uniform grid which defines the observation grid $\mathcal{X}$ via different subsampling rates. Yellow points correspond to the observation grid $\mathcal{X}$ (*In-s*) while purple points indicate off-grid points (*Out-s*). The prediction for I-MP-PDE at $t = 0$ is the interpolated initial condition.

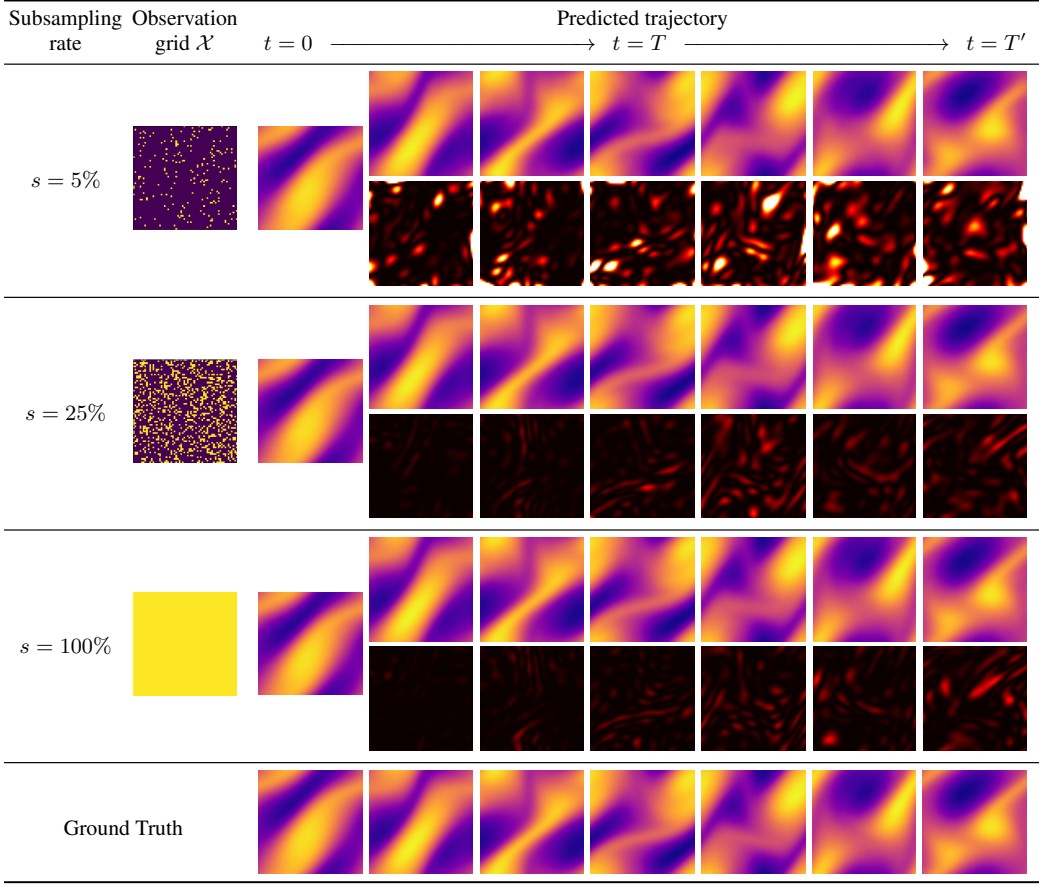

Figure 6: Prediction MSE per frame for **DINo** on *Navier-Stokes* with its corresponding observed grid $\mathcal{X}$. For each model, the first row contains the predicted trajectory from 0 to $T'$, the second row is the corresponding error maps w.r.t. the reference data (the darker the pixel, the lower the error).

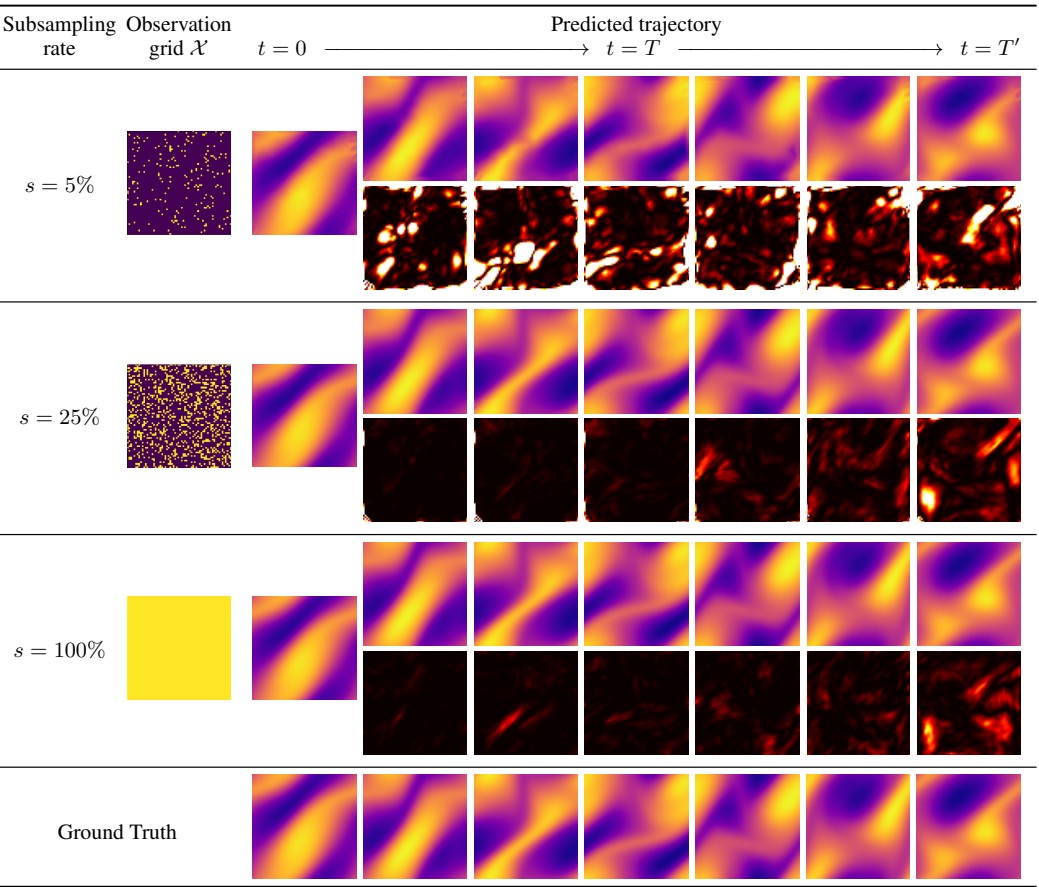

Figure 7: Prediction MSE per frame for **I-MP-PDE** on *Navier-Stokes* with its corresponding observed grid $\mathcal{X}$. For each model, the first row contains the predicted trajectory from 0 to $T'$, the second row is the corresponding error maps w.r.t. the reference data (the darker the pixel, the lower the error).

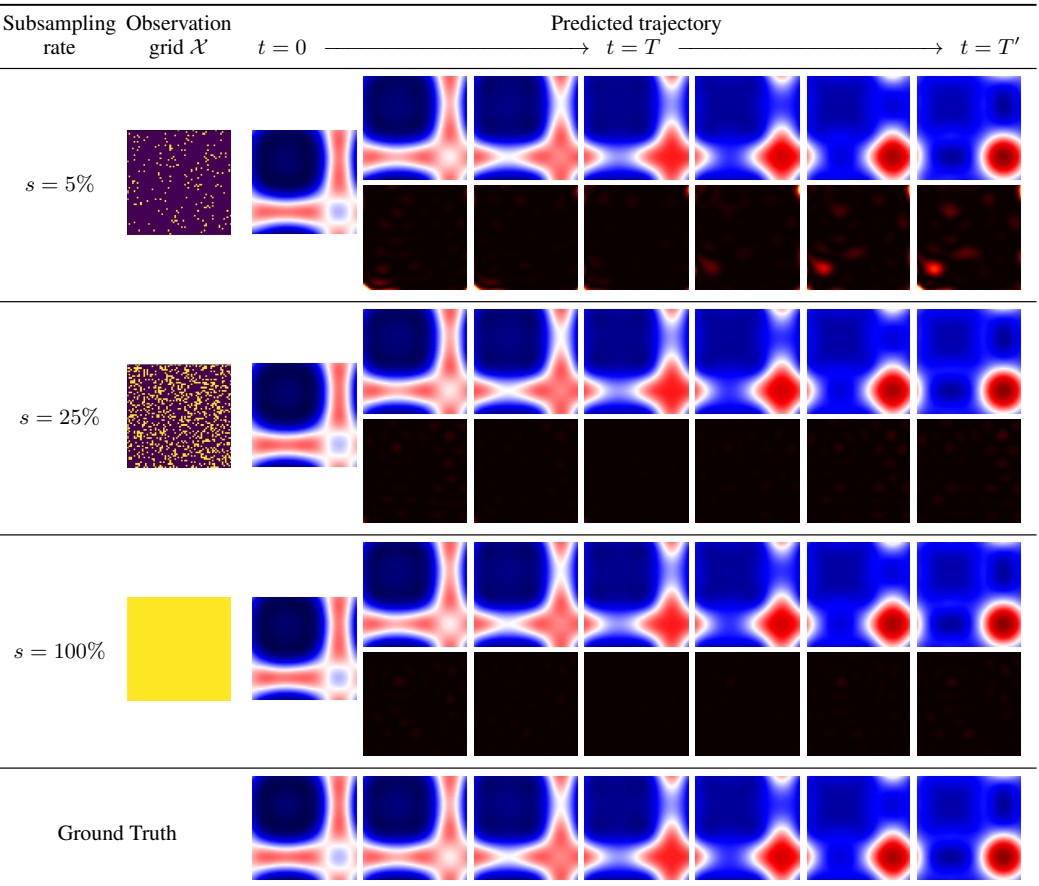

Figure 8: Prediction MSE per frame for **DINo** on *Wave* with its corresponding observed grid $\mathcal{X}$. For each model, the first row contains the predicted trajectory from 0 to $T'$, the second row is the corresponding error maps w.r.t. the reference data (the darker the pixel, the lower the error).

## C    DETAILED DESCRIPTION OF DATASETS

We choose $\mathcal{T}$ (resp. $\mathcal{T}'$) on a regular grid in $[0, T]$ (resp. $[0, T']$) with a given temporal resolution and fix $T' = 2T + \delta t$, where $\delta t$ is the step size of the temporal grid. Hence, we always consider 10 consecutive frames for *In-t* and 10 more for *Out-t*. The range of $T$ depends on the nature of the dataset. We provide below further details on the choice of these parameters and other experimental parameters, such as the number of observed trajectories.

**2D Wave equation** (*Wave*).    It is a second-order PDE:

$$\frac{\partial^2 u}{\partial t^2} = c^2 \Delta u, \tag{10}$$

where $u$ is a function of the displacement at each point in space w.r.t. the rest position, $c \in \mathbb{R}_+^*$ is the speed of wave traveling. We transform the equation to a first-order form, considering the input $v_t = \left( u_t, \frac{\partial u_t}{\partial t} \right)$, so that the dimension of $v_t(x)$ at each point $x \in \Omega$ is $n = 2$.

We generate our dataset for speed $c = 2$ with periodic boundary condition. The domain is $\Omega = [-1, 1]^2$. For initial conditions $v_0 = \left( u_0, \frac{\partial u_t}{\partial t} \big|_{t=0} \right)$, the initial displacement $u_0$ is a Gaussian function:

$$u_0(x; a, b, r) = a \exp \left( -\frac{(x - b)^2}{2r^2} \right), \tag{11}$$

where the height of the peak displacement is $a \sim \mathcal{U}(2, 4)$, the location of the peak displacement is $(b_1, b_2) \sim \mathcal{U}(-1, 1)$, and the standard deviation is $r \sim \mathcal{U}(0.25, 0.3)$. The initial time derivative is $\frac{\partial u_t}{\partial t} \big|_{t=0} = 0$. Each snapshot is generated on a uniform grid of $64 \times 64$. Each sequence is generated with fixed interval $\delta t = 0.25$. We set the train horizon $T = 2.25$ and the inference horizon $T = 4.75$. We generated 512 train trajectories and 32 test trajectories.

**2D Navier Stokes** (*Navier-Stokes*, Stokes, 1851).    This dataset corresponds to an incompressible fluid dynamics described by:

$$\frac{\partial w}{\partial t} = -u\nabla w + \nu \Delta w + f, \quad w = \nabla \times u, \quad \nabla u = 0, \tag{12}$$

where $u$ is the velocity field and $w$ the vorticity. $u, w$ lie on a spatial domain with periodic boundary conditions, $\nu$ is the viscosity and $f$ is a constant forcing term. The input $v_t$ is $w_t$ ($n = 1$). $\nu$ is the viscosity and $f$ is the constant forcing term in the domain $\Omega$.

The spatial domain is $\Omega = [-1, 1]^2$, the viscosity is $\nu = 1 \times 10^{-3}$, the forcing term is set as:

$$\forall x \in \Omega, f(x_1, x_2) = 0.1 \Big( \sin\big(2\pi(x_1 + x_2)\big) + \cos\big(2\pi(x_1 + x_2)\big) \Big). \tag{13}$$

The full spatial grid is of dimension $64 \times 64$ or $256 \times 256$ according to experiments in Section 5. We sample initial conditions as in Li et al. (2021b) to create different trajectories. The first 20 steps of the trajectories are cut off as they are too noisy and not informative in terms of dynamics. Trajectories are collected with $\delta t = 1$. We set the training horizon $T = 19$ and the inference horizon $T' = 39$. We generated 512 train trajectories and 32 test trajectories.

**3D spherical shallow water** (*Shallow-Water*, Galewsky et al., 2004).    The following problem was originally presented for numerical model testing of global shallow-water equations. They can be written as:

$$\begin{aligned} \frac{\mathrm{d}u}{\mathrm{d}t} &= -fk \times u - g\nabla h + \nu \Delta u, \\ \frac{\mathrm{d}h}{\mathrm{d}t} &= -h\nabla \cdot u + \nu \Delta h. \end{aligned} \tag{14}$$

where $\frac{\mathrm{d}}{\mathrm{d}t}$ is the material derivative, $k$ is the unit vector orthogonal to the spherical surface, $u$ is the velocity field tangent to the surface of the sphere which can be transformed into the vorticity $w = \nabla \times u$, and $h$ is the thickness of the sphere. Note that the data we observe at each time $t$ is $v_t = (w_t, h_t)$. $f, g, \nu, \Omega$ are parameters of the Earth; cf. Galewsky et al. (2004) for details.

The initial conditions are slightly modified from Galewsky et al. (2004), detailed below, to create symmetric phenomena on the northern and southern hemisphere. The initial zonal velocity $u_0$ contains two non-null symmetric bands in the both hemispheres, which are parallel to the circles of latitude. At each latitude and longitude $\phi, \theta \in [-\pi/2, \pi/2] \times [-\pi, \pi]$:

$$u_0(\phi, \theta) = \begin{cases} \left( \dfrac{u_{\max}}{e_n} \exp\left( \dfrac{1}{(\phi - \phi_0)(\phi - \phi_1)} \right), 0 \right) & \text{if } \phi \in (\phi_0, \phi_1), \\[2ex] \left( \dfrac{u_{\max}}{e_n} \exp\left( \dfrac{1}{(\phi + \phi_0)(\phi + \phi_1)} \right), 0 \right) & \text{if } \phi \in (-\phi_1, -\phi_0), \\[2ex] (0, 0) & \text{otherwise.} \end{cases} \tag{15}$$

where $u_{\max}$ is the maximum velocity, $\phi_0 = \pi/7$, $\phi_1 = \pi/2 - \phi_0$, and $e_n = \exp\left(-4/(\phi_1 - \phi_0)^2\right)$. The water height $h_0$ is initialized by solving a boundary value condition problem as in Galewsky et al. (2004). It is then perturbed by adding the following $h'_0$ to $h_0$:

$$h'_0(\phi, \theta) = \hat{h} \cos(\phi) \exp\left( -\left( \frac{\theta}{\alpha} \right)^2 \right) \left[ \exp\left( -\left( \frac{\phi_2 - \phi}{\beta} \right)^2 \right) + \exp\left( -\left( \frac{\phi_2 + \phi}{\beta} \right)^2 \right) \right]. \tag{16}$$

where $\phi_2 = \pi/4$, $\hat{h} = 120\,\text{m}$, $\alpha = 1/3$ and $\beta = 1/15$ are constants defined in Galewsky et al. (2004).

We simulate this phenomenon with Dedalus (Burns et al., 2020) on a latitude-longitude (lat-lon) grid. The size of the grid is 128 (lat) × 256 (lon). We take different initial conditions by sampling $u_{\max} \sim \mathcal{U}(60, 80)$ to generate long trajectories. These long trajectories are then sliced into shorter ones. For simulation, we take one snapshot per hour (of internal simulation time), i.e. $\delta t = 1\,\text{h}$. We stop the simulation at the 320th hour. To construct a dataset rich of dynamical phenomena, we take the snapshots within the last 160 h in a long trajectory and slice them into 8 shorter trajectoires. Also note that the data is scaled into a reasonable range: the height $h$ is scaled by a factor of $3 \times 10^3$, and the vorticity $w$ by a factor 2. In each short trajectory, $T = 9\,\text{h}$ and $T' = 19\,\text{h}$. In total, we generated 16 long trajectories (i.e. 128 short trajectories) for train, 2 for test (i.e. 16 short trajectories).

## D   IMPLEMENTATION

We provide our code at `https://github.com/mkirchmeyer/DINo`.

### D.1   ALGORITHM

We detail the algorithm of DINo for training and test via pseudo-code in Algorithm 1. Training consists in solving Eq. (6) w.r.t. $\psi, \alpha_{\mathcal{T}}, \phi$. Inference involves optimization only to find $\alpha_0$.

---

**Algorithm 1:** DINo pseudo-code

*Training:* **Input:** $\mathcal{D} = \{v_{\mathcal{T}}\}$, $\{\alpha_{\mathcal{T}}^v \leftarrow 0\}_{v \in \mathcal{D}}$, $\phi \leftarrow \phi_0$, $\psi \leftarrow \psi_0$;
**while** *not converged* **do**

    **for** $v \in \mathcal{D}$ **do** $\alpha_{\mathcal{T}}^v \leftarrow \alpha_{\mathcal{T}}^v - \eta_\alpha \nabla_{\alpha_{\mathcal{T}}^v} \ell_{\text{dec}}(\phi, \alpha_{\mathcal{T}}^v)$;            /* Modulation */

    $\phi \leftarrow \phi - \eta_\phi \nabla_\phi \left( \sum_{v \in \mathcal{D}} \ell_{\text{dec}}(\phi, \alpha_{\mathcal{T}}^v) \right)$;         /* Hypernetwork update */

    $\psi \leftarrow \psi - \eta_\psi \nabla_\psi \left( \sum_{v \in \mathcal{D}} \ell_{\text{dyn}}(\psi, \alpha_{\mathcal{T}}^v) \right)$;          /* Dynamics update */

*Test:* **Input:** $\mathcal{D}' = \{v_0\}$, $\{\alpha_0^v \leftarrow 0\}_{v \in \mathcal{D}'}$, $\phi^\star, \psi^\star, \mathcal{T}' \neq \mathcal{T}$;
**while** *not converged* **do**

    **for** $v \in \mathcal{D}'$ **do** $\alpha_0^v \leftarrow \alpha_0^v - \eta \nabla_{\alpha_0^v} \ell_{\text{dec}}(\phi^\star, \alpha_0^v)$;            /* Modulation */

**for** $v \in \mathcal{D}', t \in \mathcal{T}'$ **do**

    $\alpha_t^v \leftarrow \alpha_0^v + \int_0^t f_{\psi^\star}(\alpha_\tau^v)\, d\tau$;                /* Unroll dynamics */

    $\tilde{v}_t \leftarrow D_\phi(\alpha_t^v)$;                               /* Predict */

---

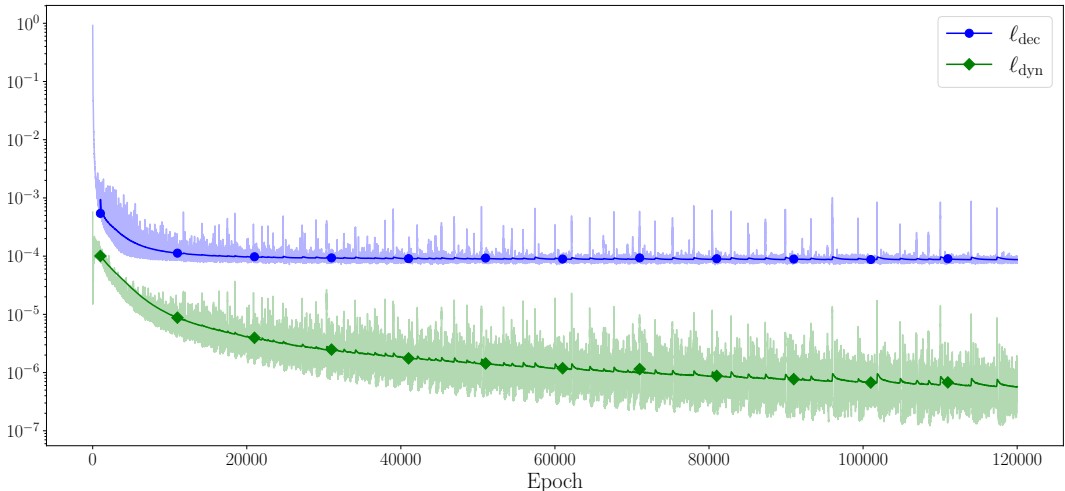

Figure 9: Learning curves on *Navier-Stokes* for $\ell_{\text{dec}}$ and $\ell_{\text{dyn}}$ throughout training (pale lines) and corresponding exponential moving averages from epoch 500 with half-life 1000 (opaque lines).

### D.2 CONVERGENCE ANALYSIS

In practice, we observe no training instability induced by the two-stage learning process of Eq. (6) and Algorithm 1: the objectives are non-conflicting. To assess this, we track the evolution of the auto-decoding loss $\ell_{\text{dec}}$ and the dynamics loss $\ell_{\text{dyn}}$ throughout training on *Navier-Stokes* ($s = 100\%$) in Figure 9. We observe that both losses smoothly converge until the end of training.

### D.3 TIME EFFICIENCY

Our auto-decoding strategy coupled with a latent neural ODE makes DINO computationally efficient compared to our best competitor MP-PDE.

**Inferring $\alpha_0$ via auto-decoding.** Given a decoder and an observation frame $v_0$, finding $\alpha_0$ corresponds to solving an inverse problem, cf. Eq. (3). At inference, we use 300 steps to infer $\alpha_0$; using less steps is possible but results in slight underfitting. This represents 2.76 s for 64 trajectories on a single Tesla V100 Nvidia GPU. Note that, as we unroll dynamics in the latent space, there is no need to relearn $\alpha_t$ when $t > 0$. Moreover, this differs from training, where $\alpha_t$ is continuously optimized for all $t \in [0, T]$ within the train horizon, in parallel with our INR decoder. Overall, we trained MP-PDE and DINO for approximately 7 days such that there is no major additional temporal training cost for DINO.

**Latent Neural ODE.** Unrolling the dynamics with a neural ODE is efficient (0.35 s for 19 time predictions for 64 trajectories on a single Tesla V100 Nvidia GPU). Indeed, the latent space is small (at most 100 dimension) and the dynamics model uses a simple four-layer MLP for $f_\psi$. With the same latent dynamics model, using an RK4 numerical scheme only incurs four additional function evaluations over a discretized alternative like a standard ResNet. This incurs a minor computational cost but enables DINO to operate at different temporal resolutions, unlike e.g. MP-PDE.

In comparison, the official code of MP-PDE takes 312 s for inference on the same hardware for the same number of trajectories (vs 3 s for DINO). MP-PDE requires building an adjacency matrix and incurs for this reason a high memory cost, especially as the number of nodes increases. Interpolation also significantly increases inference time. This is not the case for DINO, which is faster.

### D.4 ADDITIONAL IMPLEMENTATION DETAILS

We use PyTorch (Paszke et al., 2019) to implement DINO and our baselines. Hyperparameters are further defined in Appendix D.5. The dynamics model $f_\psi$ is a multilayer perceptron with swish activation function (Hendrycks & Gimpel, 2016; Ramachandran et al., 2017). Its input and output sizes are the same as the size of latent space $d_\alpha$. All hidden layers share the same size. DINO's parameters are initialized with the default initialization in PyTorch, defining $\phi_0, \psi_0, \omega$ in Algorithm 1.

Table 7: DINO's hyperparameters.

| Hyperparameter | Navier-Stokes | Wave | Shallow-water |
|---|---|---|---|
| Decoder $D_\phi = I_{h_\phi}$ | | | |
| Number of layers | 3 | 3 | 6 |
| Number of hidden channels | 64 | 64 | 256 |
| Frequency scale factor $\omega_s$ | 64 | 64 | 64 |
| Size of latent space $d_\alpha$ | 100 | 50 | 300 |
| Dynamics model $f_\psi$ | | | |
| Number of layers | 4 | 4 | 4 |
| Hidden layer size | 512 | 512 | 800 |
| Activation function | Swish | Swish | Swish |
| Optimization | | | |
| Learning rate $\eta_\phi$ | $10^{-2}$ | $10^{-2}$ | $10^{-2}$ |
| Learning rate $\eta_\alpha$ | $10^{-3}$ | $10^{-3}$ | $10^{-3}$ |
| Learning rate $\eta_\psi$ | $10^{-3}$ | $10^{-3}$ | $10^{-3}$ |
| Number of epochs | 12 000 | 12 000 | 12 000 |
| Batch size i.e. sequences per batch | 64 | 64 | 16 |

We recall that $\omega$ is fixed throughout training to reduce the number of optimized parameters without loss of performance. As in related work (Sitzmann et al., 2020; Fathony et al., 2021), the frequency parameters $\omega$ are scaled by a factor, $\omega_s$, considered as a hyperparameter. For dynamics learning, we use an RK4 integrator via TorchDiffEq (Chen et al., 2018) and apply exponential Scheduled Sampling (Bengio et al., 2015) to stabilize training. In practice, modulations $\alpha_t$ are learned channel-wise such that $I_\theta \colon \Omega \to \mathbb{R}^{d_c}$ has separate parameters per output dimension to make predictions less correlated across channels. We optimize all parameters $\phi, \alpha, \psi$ using Adam (Kingma & Ba, 2015) with decay parameters $(\beta_1, \beta_2) = (0.9, 0.999)$.

### D.5 HYPERPARAMETERS

We list the hyperparameters of DINO for each dataset in Table 7. In practice, we observe it is beneficial to decay the learning rates $\eta_\phi, \eta_\alpha$ when the loss reaches a plateau.

### D.6 BASELINES IMPLEMENTATION

We detail in the following the hyperparameters and architectures used in our experiments for the considered baselines, which we reimplemented for our paper.

- **CNODE** is implemented with four two-dimensional convolutional layers with 64 hidden features, ReLU activations, $3 \times 3$ kernel and zero padding. Learning rate is fixed to $10^{-3}$. We use an adjoint method for integration like Chen et al. (2018).

- **MNO.** We use the FNO architecture of Li et al. (2021b) with three FNO blocks, GeLU activations, 12 modes and a width of 32. Learning rate is fixed to $10^{-3}$.

- **DeepONet.** We consider an autoregressive formulation of DeepONet. We choose a width of 1000 for hidden features with a depth of 4 for both trunk and branch nets with ReLU activations. Learning rate is fixed to $10^{-5}$.

- **MP-PDE.** We adapt the implementation in Brandstetter et al. (2022) to handle two- and three-dimensional PDEs. We use a time window of 1 with pushforward trick. Batch size and number of neighbors are fixed to 8. Learning rate is fixed to $10^{-3}$. We use ReLU activations.

- **SIREN.** To represent data in space and time, SIREN takes space and time coordinates $(x, t)$ as input. To handle multiple trajectories, we concatenate an optimizable per-trajectory context code $\alpha$ to the coordinates like in DINO. We fix the hidden layer size of SIREN to 256. We initialize the parameters and use the default input scale as in Sitzmann et al. (2020). The size of the context code is $d_\alpha = 800$. The learning rate is $10^{-3}$.

Table 8: Long-term extrapolation performance of DINo and (I-)MP-PDE in the space and time generalization experiment for test trajectories on *Out-t* ($]T, T' = T + \Delta T]$); cf. Table 2 and Section 5.1.

| Subsampling ratio | Model | $\Delta T = T$ | $\Delta T = 5T$ | $\Delta T = 10T$ | $\Delta T = 50T$ |
|---|---|---|---|---|---|
| $s = 5\%$ | DINo | **2.017E−3** | **4.895E−3** | **1.209E−2** | **1.440E−1** |
| | I-MP-PDE | 8.387E−3 | 3.580E−2 | 3.356E−1 | 4.031E1 |
| $s = 100\%$ | DINo | **4.617E−4** | **2.082E−3** | **6.901E−3** | **1.215E−1** |
| | MP-PDE | 5.251E−4 | 3.524E−2 | 3.339E−1 | 9.755E1 |

Table 9: MSE reconstruction error (In-s and Out-s) of train sequences within the train horizon (In-t) for three different methods: interpolation of observed points in $\mathcal{X}_{tr}$, FourierNet learned over individual frames in $\mathcal{X}_{tr}$, and DINo (FourierNet with a dynamics model).

| MSE train In-t | Interpolation | FourierNet | DINo |
|---|---|---|---|
| Navier-Stokes, $s = 5\%$ | 8.277E−3 | **9.673E−4** | 1.029E−3 |
| Wave, $s = 5\%$ | 7.075E−4 | **4.085E−5** | 4.088E−5 |

- **MFN.** Similarly to the previous SIREN baseline, we concatenate the per-trajectory context code to space and time coordinates at the first layer. The hidden layer size is fixed to 256 and we use the default parameter initialization with a frequency scale $\omega_s$ of 64 higher than DINo. The size of the context code is $d_\alpha = 800$. The learning rate is $10^{-3}$.

The ablation "DINo (no sep.)" modulates frequencies $\omega$s through a latent shift modulation from $\alpha_t$, similarly to the bias terms $b$s in Section 4.4.

# E COMPLEMENTARY ANALYSES

We detail in this section additional experiments, allowing us to further analyze and assess the performance of DINo.

## E.1 LONG-TERM TEMPORAL EXTRAPOLATION

We provide in Table 8 an analysis of error accumulation over time for long-term extrapolation. More precisely, we generate a Navier-Stokes dataset with longer trajectories and report MSE for $T' = T + \Delta T$ where $\Delta T \in \{T, 5T, 10T, 50T\}$. Note that $\Delta T = T$ is the setting of our main experiments ($T' = 2T$).

We observe that DINo's MSE in long-term forecasting is more than an order of magnitude smaller than for (I-)MP-PDE. This demonstrates the extrapolation abilities of our model.

## E.2 INRS' ADVANTAGE OVER INTERPOLATION

We report in Table 9 the MSE of bicubic interpolation, our FourierNet's MSE (auto-decoding with amplitude modulation but without dynamics model) and DINo's MSE (with dynamics model) on train *In-t* for both *Navier-Stokes* and *Wave*. This corresponds to MSE averaged over all training frames within the train horizon and not only the initial condition $v_0$.

We observe that FourierNet is better than interpolation. Indeed, interpolation is poorly adapted to sparse observation grids: the interpolation errors are clearly visible in Figure 7, first row (5% setting). Interestingly, DINo's MSE is only slightly worse than the FourierNet's MSE, showing that we correctly learned the dynamics of latent modulations $\alpha_t$. I-MP-PDE, which combines bicubic interpolation with MP-PDE, is then expectedly outperformed by DINo on this challenging 5% setting. This shows the advantage of using INRs instead of standard bicubic interpolation to interpolate between observed spatial locations.

Table 10: SST test prediction performance for DINo and VarSep (Donà et al., 2021).

| Method | MSE |
|--------|-----|
| VarSep | 1.43 |
| DINo | **1.27** |

### E.3 Modeling real-world data

**SST.**    We evaluate DINo on real-world data to further assess its applicability. Following de Bézenac et al. (2018) and Donà et al. (2021), we model the Sea Surface Temperature (SST) of the Atlantic ocean, derived from the data-assimilation engine NEMO (Nucleus for European Modeling of the Ocean, Madec & NEMO System Team) using E.U. Copernicus Marine Service Information.[1] Accurately modeling SST dynamics is critical in weather forecasting or planning of coastal activities. This problem is particularly challenging as SST dynamics are only partially observed: several unobserved variables affecting the dynamics (e.g. the sea water flow) need to be estimated from data.

For this experiment, we consider trajectories collected from three geographical zones (17 to 20) following the initial train / test split of de Bézenac et al. (2018). Notably, $T = 9\,\mathrm{d}$, which includes $\tau = 4\,\mathrm{d}$ of conditioning frames, i.e. models are tested to predict $v_{t\in[\![\tau,T]\!]}$ from $v_{t\in[\![0,\tau-1]\!]}$.

**Incorporating consecutive time steps.**    To model SST which includes non-Markovian data and thus does not correspond to an Initial Value Problem as in Section 2, we modify our dynamics model in a similar fashion to Yıldız et al. (2019) to integrate a history of several consecutive observations $v_{t\in[\![0,\tau-1]\!]}$ instead of only the initial observation $v_0$. In more details, we define a neural ODE over an augmented state $[\alpha_t, \alpha'_t]$ where $\alpha_t$ is our auto-decoded state and $\alpha'_t$ is an encoding of $\tau = 4$ past auto-decoded observations via a neural network $c_\xi$. We adjust our inference and training settings as follows:

- inference: we compute $\alpha'_{\tau-1} = c_\xi(\alpha_0, \ldots, \alpha_{\tau-1})$ and then unroll our neural ODE from the initial condition $[\alpha_{\tau-1}, \alpha'_{\tau-1}]$ to obtain $[\alpha_t, \alpha'_t]$ for all $t > \tau - 1$:

$$\forall t \in [\![0, \tau-1]\!], \alpha_t = e_\varphi(v_t), \quad \alpha'_{\tau-1} = c_\xi(\alpha_0, \ldots, \alpha_{\tau-1}), \quad \frac{\mathrm{d}[\alpha_t, \alpha'_t]}{\mathrm{d}t} = f_\psi([\alpha_t, \alpha'_t]);$$

- training: for all $t$, we infer $\alpha'_{t+\tau-1} = c_\xi(\alpha_t, \ldots, \alpha_{t+\tau-1})$ and fit the above neural ODE on the $[\alpha_t, \alpha'_t]$ obtained for all $t \in [\![0, T-\tau+1]\!]$.

This experiment confirms that our space- and time-continuous framework can easily be extended to incorporate refined temporal models.

**Results.**    We report in Table 10 test MSE for DINo and VarSep (Donà et al., 2021), the current state-of-the-art on SST, retrained on the same training data. DINo notably outperforms VarSep in prediction performance. This demonstrates DINo's potential to handle complex real-world spatiotemporal dynamics. We also provide some visualizations of DINo's train and test predictions in Figure 10. We make two observations. First, DINo fits very accurately the train data. Second, on the test data, we observe that the dynamics on low frequencies seem to be correctly modeled while the prediction of high frequencies dynamics are less accurate. Larger scale experiments would be required to effectively evaluate the model performance on this challenging dataset. Given the complexity of the data, this is out of the scope of the paper. Yet, these experiments already demonstrate that DINo behaves competitively w.r.t. the previous state-of-the-art.

**Implementation choices.**    We choose a similar INR and dynamics architecture as for our Shallow-water experiment. We use for $c_\xi$, which takes as input four consecutive $\alpha_t$s, individual encoding of the $\alpha_t$s through a four-layer fully connected network which are then fed to a single linear layer.

---

[1]https://data.marine.copernicus.eu/product/GLOBAL_ANALYSIS_FORECAST_PHY_001_024/description.

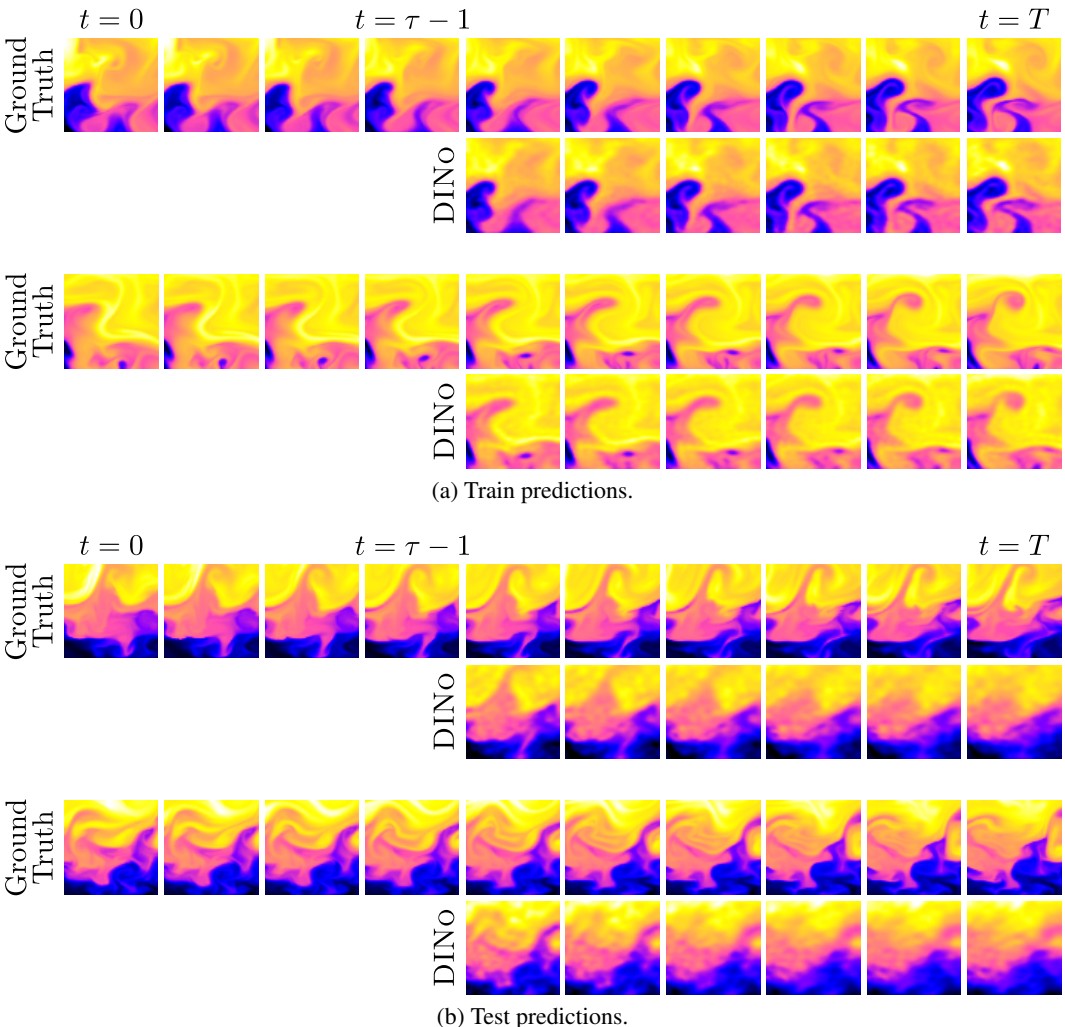

(a) Train predictions.

(b) Test predictions.

Figure 10: DINO's prediction examples on SST.

