# OpenReview forum: "Continuous PDE Dynamics Forecasting with Implicit Neural Representations"
_ICLR.cc/2023/Conference — ICLR 2023 notable top 25%_

### Official Review · Reviewer_4pqH · 2022-10-19

**Confidence:** 3
**Correctness:** 3
**Technical Novelty And Significance:** 4
**Empirical Novelty And Significance:** 4
**Recommendation:** 8

**Clarity, Quality, Novelty And Reproducibility:**

The paper is well written and seems novel. Reproducibility is possible as the authors released the code. Yet, some additional details could be provided in the appendix.

**Strength And Weaknesses:**

I really enjoyed this paper. Forecasting solutions of PDEs in a continuous manner is a very interesting topic yet very challenging. Such techniques may alleviate the need of heaving meshing procedure in standard physics solver. The proposed approach seems reasonable: while using INR for modeling continuous solutions seems the natural thing to do, conditioning its parameters to the initial condition is not straightforward. Amplitude modulation seems like a very clever and interesting method to tackle the problem. However, I may need additional precisions to fully understand the contribution.

- I do understand the advantages of using FourierNet with the chosen architecture for the hyper-network in theory. However, I wonder if such gain is verified in practice. For example, could it be possible to replace FourierNet by SIREN or MFN conditioned with an hyper-network (and not directly through the input) ? Could the authors briefly discuss about the other structure they tried ? I have the same kind of question for the auto-decoder: is it actually beneficial over an explicit learned encoder model ?

- I am not absolutely convinced by the "separation of variable" argument in section 4.4. It is not clear to me how separation of variable is relevant in this situation. This technique allows to disentangle a PDE in a set of ODEs, which are easier to solve. Yet, in this framework, the PDE seems to be directly translated into a ODE in the latent space $\alpha_t$, where the dynamics solely depends on the time derivatives. I would appreciate if the authors could clarify this statement.

- Related to my previous remark, I am not sure how the space derivatives (which are needed to solve the PDE) can be embedded in the latent vector $\alpha_t$ such that the entire dynamics can be resumed as a temporal ODE. Does the authors have any insights about what is happening here ? May the auto-decoder somehow pre-computes spatial derivatives (even if the observation is partial) and embed them in a convenient form in $\alpha_t$ ?

I also acknowledge the release of the code, which actually helps me to answer to more technical question about the model (especially, how the partial observation is inputted to the model). However, I think that appendix section D could be more precise.

My most important questions focuses on the experiments. I do appreciate the comprehensive study conducted by the authors, with relevant and numerous baselines, and extensive analysis on the generalization capacity of DINo. Yet, I am still a bit skeptical about the chosen datasets, for three reasons, ordered below in ascending order of importance:

- The main results (table 2 and 3) are computed on uniform grids. I do not see any aspect of DINo that are not compatible with uniform meshes, which are much more powerful and widely used to solve PDEs, especially Navier-Stokes equations. However, uniform grid contains a lot of regularities that both DINo and the baselines may leveraged. I wonder how the proposed method would behave on irregular triangular mesh with complex physics such as the ones introduced in [1].

- Both qualitatives and quantitatives results may indicates that the task on which DINo is evaluated are actually saturated. My doubts are based on the very close MSE obtained in the standard setup $s=100$% and the merely indistinguishable difference between figure 6 and 7. This hypothesis could quickly be rejected by showing additional prediction example from both DINo and I-MP-PDE, maybe with some failure cases. This seems important: if the tasks are saturated, it is unclear how such tasks are relevant for assessing DINo generalization capacities.

- I was quite amazed by the performances of DINo in the 5% subsampling scenario. According to me, these results, and actually many others, may be explained by one the three following hypothesis:
     1. DINo can leverage knowledge for the PDE dynamics to extrapolate un-observed information.
     2. DINo exploits biaises in the data to infer the state in the entire space.
     3. Datasets are not sufficiently complex and basic interpolation is enough.
In my opinion, (3) is highly possible, as I-MP-PDE seems to perform competitively to DINo in the 5% setup. (3) may be easily rejected by comparing the reconstruction error of DINo on $v_0$ with direct bi-cubic interpolation from the partial observations. If the error on the latter is bigger than the error on the former, then DINo does leverages knowledge either from physics or from regularities in the dataset. Checking the validity of (2) is much more challenging. Maybe the easiest way could be to look at a wider set of trajectories from the dataset and make sure to observe sensible diversity among them. I would really appreciate if the authors could provide an in-depth discussion concerning these points.

I also wonder why I-MP-PDE is absent from table 3, it seems to me that it could be compared with DINo in this setup as well.


[1] Tobias Pfaff, Meire Fortunato, Alvaro Sanchez-Gonzalez, and Peter W. Battaglia. Learning mesh-
based simulation with graph networks. In International Conference on Learning Representations
(ICLR), 2021

**Summary Of The Paper:**

This paper introduces DINo, a continuous time and space solver for partial differential equations. The authors proposed to model the solution of the PDE as an implicit neural representation conditioned on the initial condition through an hyper-network. The dynamics is modeled as a learned latent ordinary differential equation.

**Summary Of The Review:**

I really enjoyed the paper and find the overall model architecture quite clever. However, I still have some doubts about the performances of the model compared to more trivial interpolation techniques combined with existing methods. The paper provides a quite extensive comparison with SOTA, yet I am not sure that the chosen task are actually relevant to correctly assessed the model. If the authors provides satisfying answers to my questions, I will be very pleased to increase my grade.

## EDIT
The authors provided very detailed and convincing answers, which remove my fears on this paper. I now think that this work deserves to be presented at ICLR. I still have some concern about the practical performance of the model. Although the datasets are perfectly within the standards of the community, these tasks are already relatively well mastered. I would have given the maximum score if the technical contribution (definitely respectable) had been completed with new results on more difficult datasets than those conventionally used in the state of the art.

It is nevertheless a work of excellent quality, which opens the door to new approaches for the resolution of PDE, and which, I believe, deserves the attention of the community. I increased my rating from 6 to 8.

---

> ### Author Response · Authors · 2022-11-14
> **Response to Reviewer 4pqH - 1/3**
>
> We thank the reviewer for the thoughtful review. We address the concerns below. We will complement it later with a revision of our paper.
>
> ## Architecture choices
>
> ### Alternative INR Architecture
>
> We implemented DINo using FourierNets, a special case of MFNs. Other INR alternatives such as SIREN are possible. For example, [1] proposed, as we do, a simple hypernetwork-based approach tailored to SIREN for static problems.
>
> We chose FourierNets for two reasons. First, it is state-of-the-art; Table 2 shows that MFN/FourierNet outperform SIREN within the train horizon where spatiotemporal INRs are well defined. Second, it is interpretable as it defines a simple relation between its parameters on one hand and the spatial basis and amplitudes on the other, unlike SIREN [2]. Given this choice, we designed a FourierNet modulation to perform space-time variable separation. To this end, we only modulate the amplitudes of the spatial basis via a hypernetwork. We demonstrated in Table 2 that separation of variables significantly improved performance, as evidenced by the gain over DINo (no sep.), which does not perform separation of variables. Note that it is unclear how to efficiently perform separation of variable with SIREN.
>
> [1] Dupont et al. From data to functa: Your data point is a function and you can treat it like one. ICML 2022.\
> [2] Yüce et al. A Structured Dictionary Perspective on Implicit Neural Representations. CVPR 2022.
>
> ### Encoder vs. Auto-Decoding
>
> It is possible to replace auto-decoding with standard auto-encoding. We chose auto-decoders as they are simple to implement and efficient. They remove the need for defining flexible encoders applicable to non-planar geometries or irregular grids and for tuning hyperparameters and architectures. A standard auto-encoding choice would be to use set encoders as in neural processes [3]; yet, they were shown to underfit [4].
>
> [3] Garnelo et al. Conditional Neural Processes. ICML 2018.\
> [4] Kim et al. Attentive neural processes. ICML 2019.
>
> ## Separation of Variables Interpretation
>
> We follow a fully data-driven approach for modeling observed data which follow an unknown PDE. Thus, we do not refer to separation of variables as a *PDE resolution technique* for known PDEs; we refer to it as a *modeling assumption*. Note that this assumption is also used in eigenfunction expansion-based methods [5].
>
> Separation of variables parameterizes our solution via an amplitude-modulated FourierNet (Eq. (8)) which constrains:
> $$\tilde v_{t}(x) = \sum_{m} c^{(m)}(t) \cdot s_{\gamma^{(m)}}(x) + \text{cst}.$$ We empirically assessed its validity on our datasets (cf. the ablation of DINo (no sep.) in Section 5.2, Table 2). We will clarify Section 4.4 in our revision to explicitly present separation of variables as a modeling assumption.
>
> [5] Kutz. Data-driven modeling \& scientific computation. Oxford University Press, 2013.
>
> ## Encoding of Spatial Derivatives in the Latent Space
>
> We clarify that DINo is a purely data-driven approach which aims at modeling solutions given by an unknown PDE. The underlying PDE being unknown, we do not explicitly compute the spatial derivatives. Instead, they are implicitly embedded into $f_\psi$, the dynamics model of $\alpha_t$.
>
> To illustrate this, let us assume that our data follows a PDE of the form $\frac{\partial v}{\partial t} = N(v, \frac{\partial v}{\partial x} , \frac{\partial^2 v}{\partial x^2}, \dots, x, t;\beta)$, where $N$ describes the nonlinear evolution. Suppose that the time-space solution takes the form of $v(x, t)=\sum_{m=1}^M c_m(t) s_m(x)$ (separation of variables assumption). By replacing it into the PDE above, it is easy to show that the solution satisfies an ODE: $$ \forall k~~ \sum_m \left\langle s_m, s_{k}\right\rangle\frac{d c_m}{d t}=\left\langle N\left(\sum_m c_m s_m, \sum_m c_m\frac{\partial s_m}{\partial x}, \sum_m c_m\frac{\partial^2 s_m}{\partial x^2}, \cdots, x, t, \beta\right), s_{k}\right\rangle,$$ from Eq. (11.10), p.377 in [6]. Spatial transformations now only affect the basis functions ($s_k$) such that the PDE is transformed into a purely temporal ODE on $c_m$. Our solution takes this form in Eq (8): our INR models the spatial basis $\{s_m\}$ and $\alpha_t$ models its amplitudes $\{c_m\}$. If we knew beforehand the PDE, we could solve explicitly the ODE on the amplitudes which involves the spatial derivatives of the spatial basis. This is not our case. Therefore, our neural ODE accounts for the temporal dynamics of the coefficients and implicitly involves these spatial derivatives.
>
> [6] Brunton and Kutz. Data-Driven Science and Engineering: Machine Learning, Dynamical Systems, and Control. Cambridge, 2019.
>
> ## Implementation Details (Appendix D)
>
> Partial observations are injected into the model via Eq. (6). We will add some more details in our revision to clarify Appendix D.

---

> > ### Author Response · Authors · 2022-11-14
> > **Response to Reviewer 4pqH - 2/3**
> >
> > ## Experiments
> >
> > ### Uniform Meshes
> >
> > We clarify our experiments in Table 2 and 3 as this might have been unclear. Except for scenario $s=100$\%, training is performed on non-uniform observation grids. We display our observed spatial locations in Figures 6 and 7 (second column) in the Appendix. Regarding evaluation, we proceed in two different ways in Table 2 and 3:
> >
> > * In Table 2, evaluation is performed on new unseen points on a uniform grid. This is only the case for evaluation and is not restrictive as adaptation to other grid choices is direct.
> > * In Table 3, it is performed on a new non-uniform grid except for $s=100$\%.
> >
> > In short, DINo can indeed handle non-uniform grids and our experiments in Section 5.2 all consider non-uniform grids. Moreover, we stress that DINo is fully meshless unlike [7].
> >
> > [7] Pfaff et al. Learning mesh-based simulation with graph networks. ICLR 2021.
> >
> > ### Saturated datasets
> >
> > We demonstrate that our datasets, under our evaluation scenarios, are not saturated.
> >
> > * *Superresolution failures.* Visually, we clearly see on these videos for [Navier-Stokes (*link*)](https://anonymous.4open.science/r/DINo-D752/visu/navier_stokes.gif) and [Shallow Water (*link*)](https://anonymous.4open.science/r/DINo-D752/visu/shallow_water.gif) some failure cases in the superresolution setting. On Navier-Stokes, MP-PDE simply fails, while on Shallow-Water DINo starts accumulating some errors when extrapolating out of the train time horizon ($t>T$). DINo is a first step towards accurately handling this challenging setting.
> >
> > * *Out-t failures.* When resolution is fixed, DINo's and MP-PDE's MSE are close as they are state-of-the-art. Yet, when looking at the Out-t MSE, we note that the gap between MP-PDE and DINo increases, especially when we increase the test horizon as in the table below where we vary $\Delta T\in\\{T, 5T, 10T, 50T\\}$ in $T'=T+\Delta T$.
> >
> > | Out-t i.e. $]T, T+\Delta T]$ | Model | $\Delta T=T$ | $\Delta T=5T$ | $\Delta T=10T$ | $\Delta T=50T$ |
> > |--------| --------   | -------- | --------   |--------   | --------   |
> > | Navier-Stokes, $s = 5$\% | DINo  | **1.813e-3** | **4.895e-3** | **1.209e-2** | **1.440e-1**|
> > |                   | I-MP-PDE| 8.225e-3 | 3.580e-2 | 3.356e-1 | 4.031e+1 |
> > | Navier-Stokes, $s = 100$\% | DINo  | **4.311e-4** | **2.082e-3** | **6.901e-3** | **1.215e-1** |
> > |                   | MP-PDE| 5.856e-4 | 3.524e-2 | 3.339e-1 | 9.755e+1 |
> >
> > * *Investigation of per trajectory performance.* Finally, we propose to further investigate the error (in-t+out-t) per test trajectory in the evaluation setting of Table 2. We consider another metric, Mean Average Percentage Error (MAPE) in \% defined as $100\frac{\Vert v_{true}-v_{predicted}\Vert_2}{\Vert v_{true}\Vert_2}$ and report it below per test trajectory (MAPE/traj) for DINo and MP-PDE. We consider our original test set and a new set with 10 times more trajectories drawn from the same distribution.
> >
> > | Navier-Stokes, $s=100$\% | MAPE/traj mean | MAPE/traj stddev | MAPE/traj min | MAPE/traj max |
> > |--------| --------   | --------   |--------   | --------   |
> > | DINo | 1.841\% | 0.417\% | 1.323\% | 2.986\% |
> > | DINo (10x more test trajectories) | 1.967\% | 0.568\% | 1.106\% | 5.475\% |
> > | MP-PDE | 2.088\% | 0.351\% | 1.531\% | 2.945\% |
> > | MP-PDE (10x more test trajectories) | 2.146\% | 0.515\% | 1.176\% | 4.865\% |
> >
> > We make two observations on the latter table. First, there is some variety per trajectory: MAPE/traj reaches up to 5\% and can go as low as 1\%. This error might be hardly perceptible to the naked eye, yet appears in the heatmaps reported in Appendix Figures 6 and 7. Second, these scores are similar to those reported in related work. FNO [8] reports a MAPE of $\sim$1\% on all their datasets within the train horizon; we report performance of $\sim$2\% within and beyond the train horizon (In-t+Out-t). In the $s=5$\% setting, this performance reaches $\sim$4\% (as shown below). Therefore, our datasets have similar or higher difficulty than recent papers in the domain.
> >
> > [8] Li et al. Fourier Neural Operator For Parametric Partial Differential Equation. ICLR 2021.

---

> > > ### Author Response · Authors · 2022-11-14
> > > **Response to Reviewer 4pqH - 3/3**
> > >
> > > ### Performance in the 5\% setting
> > >
> > > We attempt to show that hypothesis 1. is probably the best explanation for DINo's performance in the challenging 5\% setting. To this end, we infirm hypothesis 2. and 3. with the recommended additional studies.
> > >
> > > **1. DINo can leverage knowledge for the PDE dynamics to extrapolate un-observed information.**
> > >
> > > DINo does not use explicit knowledge of the (unknown) PDE. Yet, it introduces two key components: (i) better time extrapolation via a neural ODE and (ii) better spatial modeling by sharing spatial information across frames via an INR decoder. These two key components are the best explanation for DINo's performance in the 5\% setting. They provide strong inductive biases allowing DINo to learn the true dynamics from the data only without knowledge of the underlying PDE.
> > >
> > > **2. DINo exploits biases in the data to infer the state in the entire space.**
> > >
> > > As recommended, we consider a larger number of test trajectories (5 and 10 times the number in our original submission). We display DINo's MAPE per trajectory, which captures the difficulty of modeling each trajectory, hence a notion close to diversity. We observe that the mean MAPE does not change as we increase the number of trajectories: DINo is not biased by our original test set. Note that if the test data were biased, then all the models could equally benefit from this bias. Yet, DINo clearly outperforms alternatives in this setting: it is 8 times better than I-MP-PDE on Navier-Stokes. Moreover, we stress that our datasets are standard and also used in related work e.g. [8] such that any potential bias would also concern these work.
> > >
> > > | Navier-Stokes, $s=5$\% | MAPE/traj mean | MAPE/traj stddev | MAPE/traj min | MAPE/traj max |
> > > |--------| --------   | --------   |--------   | --------   |
> > > | 64 trajectories (original)  | 4.189\% | 0.362\% | 3.567\% | 5.029\% |
> > > | 320 trajectories (x5)  | 4.396\% | 0.664\% | 3.388\% | 8.230\% |
> > > | 640 trajectories (x10)  | 4.392\% | 0.674\% | 3.304\% | 8.230\% |
> > >
> > > [8] Li et al. Fourier Neural Operator For Parametric Partial Differential Equation. ICLR 2021.
> > >
> > > **3. Basic interpolation is enough**
> > >
> > > We report below the MSE of bicubic interpolation, our FourierNet's MSE (auto-decoding with amplitude modulation but without dynamics model) and DINo's MSE (with dynamics model) on train In-t for both Navier-Stokes and Wave. This corresponds to MSE averaged over all training frames within the train horizon and not only the initial condition $v_0$.
> > >
> > > | MSE train In-t | Interpolation | FourierNet | DINo |
> > > |--------| --------   | --------   |--------   |
> > > | Navier-Stokes, $s=5$\%   | 8.277e-3      | **9.673e-4**|1.029e-3      |
> > > | Wave, $s=5$\%   | 7.075e-4      | **4.085e-5** |4.088e-5      |
> > >
> > > We observe that FourierNet is better than interpolation. Indeed, interpolation is poorly adapted to sparse observation grids: the interpolation errors are clearly visible in Figure 7 in the Appendix, first row (5\% setting). Interestingly, DINo's MSE is only slightly worse than the FourierNet's MSE, showing that we correctly learned the dynamics of latent modulations $\alpha_t$. I-MP-PDE, which combines bicubic interpolation with MP-PDE, is then expectedly outperformed by DINo on this challenging 5\% setting.
> > >
> > > ## I-MP-PDE in Table 3
> > >
> > > In Table 3, our inference grid is fixed to the observation grid (Section 5.1, paragraphs **Tasks** / **Flexibility w.r.t. input grid**). In this case, interpolation is not needed. We will clarify this in our revision. Note that this setting benefits GNNs such as MP-PDE the most as it removes interpolation error which is high in the challenging 5\% setting. Yet, even in this setting, DINo outperforms MP-PDE.

---

> > > > ### Author Response · Authors · 2022-11-18
> > > > **Second Response to Reviewer 4pqH: Additional Real-World Experiment**
> > > >
> > > > To complement our analysis of DINo on simulated physical data, we propose to evaluate DINo on non-simulated real-world data to dissipate any doubts on its applicability to more complicated problems. We provide some further details on this real-world experiment in Appendix Section E.3 of the revision and in the [response to Reviewers swAV and eLsK (link)](https://openreview.net/forum?id=B73niNjbPs&noteId=iiL2Fl_9K3). We would be happy to answer any further question.

---

> > > > > ### Comment · Reviewer_4pqH · 2022-11-21
> > > > > **About variable separation**
> > > > >
> > > > > Thank you for your very complete and very convincing answer. I have increased my rating by two. While you convinced me that your tasks are not saturated, I still believe that they are too simple for current SOTA, since the discrepancies you found required to look in very convoluted setups, either with extremely long simulations or a very large number of testing trajectories. I think that moving on to more challenging tasks should be the next objective for the community, but I understand that this is not the purpose of this paper.

---

> > > > > > ### Author Response · Authors · 2022-11-21
> > > > > > **Thank you**
> > > > > >
> > > > > > We thank the reviewer for carefully reading our answer and increasing the score. We agree that using more complicated datasets and introducing standard benchmarks is an important future objective for the PDE modeling community. Note that this is not a trivial step: in our group we made different attempts for moving from principled ML methods to more applied settings and this always requires additional engineering work.
> > > > > >
> > > > > > In an effort to answer your question, we made some experiments on a more complex dataset built from real data, aimed at predicting the dynamics of the Sea Surface Temperature (SST) of the Atlantic ocean [(link)](https://data.marine.copernicus.eu/product/GLOBAL_ANALYSIS_FORECAST_PHY_001_024/description). On this more challenging task, we were able to outperform a recent state-of-the-art neural spatiotemporal forecaster.
> > > > > > You may find additional details in our revision (Appendix Section E.3).

---

### Official Review · Reviewer_eLsK · 2022-10-22

**Confidence:** 4
**Correctness:** 4
**Technical Novelty And Significance:** 4
**Empirical Novelty And Significance:** 3
**Recommendation:** 8

**Clarity, Quality, Novelty And Reproducibility:**

- as mentioned the paper is clearly written and of high quality
- as far as I know the detailed approach is novel
- a link to PyTorch code is provided, and details are described, making the model reproducible


**Strength And Weaknesses:**

trengths.
- the paper is very well written, and easy to understand.
- the topic of continuous-time, space-continous modeling is theoretically interesting, and offers more generality than standard machine learning approaches on a grid - hence, is very interesting to the community in my opinion.
- a novel method is worked out, which makes intutive sense. In particalar the proposed separation of time & space, termed amplitude modulation, seems to improve performance.

Weakenesses.
- all datasets are somewhat synthetic. This offers a lot of possiblity to investigate the method (which the authors do), but showing performance on a real-world dataset would significantly strengthen the paper (e.g. on a weather foracasting task, or on a multi-variate time-series task, e.g. from e-Commerce).
- limitations of the method should be worked out clearer: e.g. if I understand correctly, the method cannot easily integrate information from observing several time-points? Another one: the MSE loss on the latents implies uni-modality (a single Gaussian), and determinism - can the model handle multi-modalities and stochasticities in the forecasts?

Small things:
- in several places you write 'extrapolation' (e.g. abstract, p.2. ...). However, while the method can perform extrapolation well in time, due to the ODE, it probably performs not well on extrapolation in space. The case of changing the grid to points within the convex hull of seen ones (e.g. superresolution etc.), I'd consider to be an interpolation task. Hence, please use 'interpolation' for the spatial tasks in the paper.
- Figure 4: from eq. 9 it seems b and \mu are similar - there should be also a plut between W and b then. Also, the z_ts point to b, while in the formula only W is modulated by z.

UPDATE:
Thanks for the changes made to the paper, which nicely removes some of the limitations, and adds a real-world dataset (although only in the appendix). I upgraded the score accordingly.


**Summary Of The Paper:**

The paper "Continuous PDE Dynamics Forecasting with Implicit Neural Representations" presents a space and time continuous model for forecasting. This is achieved by coupling an autodecoder to estimate a latent state, a hypernetwork-based emmission model that parameterizes an implicit neural network, and by modeling the devleopment of the latent via a learned flow with an ODE. The method, termed DINo, is compared to baselines on three datasets (2D wave equation, 2D Navier Stokes, and 3D spherical shallow water), and shown to outperform other methods in the flexible setting of changing sampling grids.


**Summary Of The Review:**

The paper is well written, and proposes a novel approach in an interesting domain. The biggest weakness is the lack of comparison on a real-world task.

---

> ### Author Response · Authors · 2022-11-14
> **Response to Reviewer eLsK**
>
> We thank the reviewer for the thoughtful review. We address the concerns below. We will complement later with a revision of our paper.
>
> ### Real-World Dataset
>
> Our PDE datasets all correspond to simulation of complex spatiotemporal systems involved in many real-world applications (e.g. atmosphere modeling or weather forecasting). Simulated data is currently standard for benchmarking data-driven PDE forecasters in the ML community. Our datasets have *similar or higher difficulty* than recent papers at top-tier ML conferences, e.g. [1,2,3,4]. Note that our Shallow-Water spherical PDE is not trivial to model without specialized encoder architectures and is relevant for modeling phenomena on the Earth; to the best of our knowledge, it is modeled *for the first time* with a fully data-driven approach. If time permits, we will attempt to evaluate DINo on data collected in a real-world setting, as recommended by the reviewer.
>
> [1] Li et al. Fourier Neural Operator For Parametric Partial Differential Equation. ICLR 2021.\
> [2] Yin et al. Augmenting Physical Models with Deep Networks for Complex Dynamics Forecasting. ICLR 2021 (oral).\
> [3] Brandstetter et al. Message Passing Neural PDE Solver. ICLR 2022 (spotlight).\
> [4] Boussif et al. MAgNet: Mesh Agnostic Neural PDE Solver. NeurIPS 2022.
>
> ### Limitations Discussion and Possible Extensions
>
> We thank the reviewer for raising these interesting points. We recall our assumptions in the following, which we will clarify in our revision (in Sections 2 and 4).
>
> * We aim at modeling Initial Value Problems (Section 2), where the knowledge of $v_t$ at any time $t$ suffices to infer $v_{t'}$ at any other time $t'>t$. Hence, we did not consider integrating several time frames.
> * We consider only deterministic PDEs. Handling stochastic PDEs and multimodality would require a combination of the extensions proposed by the reviewer and are an exciting future direction.
>
> Our main contribution consists in combining a spatial INR (auto-)decoder with a dynamics model. Extensions of the temporal model as suggested can be integrated into our framework.
>
> ### Spatial Interpolation vs. Extrapolation
>
> By extrapolation, we mean producing a prediction at a location unseen at training time. We agree that the current terminology is ambiguous and will therefore refer to spatial interpolation in our upcoming revision to avoid any confusion. Note however that INRs have also demonstrated impressive spatial extrapolation capabilities even outside the image boundaries [5], i.e. in our context outside the spatial domain $\Omega$.
>
> [5] Skorokhodov et al. Adversarial Generation of Continuous Images. CVPR 2022.
>
> ### Figure 4 (Illustration of Amplitude Modulation)
>
> We thank the reviewer for helping us to improve this figure, which we will update in our revision as shown [here (*link*)](https://anonymous.4open.science/r/DINo-D752/visu/modulation-2.pdf); we will also further detail the caption to improve clarity. We meant to represent a linear layer at each level. The weight matrix $W$ is now represented on the left and the bias term $b+\mu_t$ on the right. The input $z_t^{(l)}$ at each layer is taken as input to this linear layer and not only to $b$.

---

> > ### Author Response · Authors · 2022-11-18
> > **Second Response to Reviewer eLsK: Additional Real-World Experiment and Model Extension**
> >
> > As recommended by the reviewer, we include the following changes in our revision. We would be happy to answer any further question.
> >
> > **Real-world data.** We evaluate DINo on real-world data. Following [1,2], we model the Sea Surface Temperature (SST) of the Atlantic ocean, derived from the data-assimilation engine [NEMO (Nucleus for European Modeling of the Ocean) (link)](https://data.marine.copernicus.eu/product/GLOBAL_ANALYSIS_FORECAST_PHY_001_024/description) using E.U. Copernicus Marine Service Information [3]. Accurately modeling SST dynamics is critical in weather forecasting or planning of coastal activities. This problem is particularly challenging as SST dynamics are only partially observed: several unobserved variables affecting the dynamics (e.g. the sea water flow) need to be estimated from data.
> >
> > **Integrating information from several time points.** To model SST which includes non-Markovian data, we modify our dynamics model in a similar fashion to [4] to integrate a history of several consecutive observations. This experiment confirms that our space- and time-continuous framework can easily be extended to incorporate other rich temporal models. In more details, we define a neural ODE over an augmented state $[\alpha_t, \alpha_t']$ where $\alpha_t$ is our auto-decoded state and $\alpha'_t$ is an encoding of $\tau=4$ past auto-decoded observations via a NN $c$. We adjust our inference and training settings as in Appendix Section E.3 of the revision.
> >
> > **Results.**
> > For this experiment, we train DINo on trajectories collected from three geographical zones (17 to 20) and fix $\tau=4$ and $T=9$ days. We report MSE for test trajectories within the time interval $[\tau, T]$ for DINo and VarSep [2], the current state-of-the-art on SST, adapted to our inference procedure and retrained on the same training data.
> >
> > | Method  | MSE |
> > |--------| --------   |
> > | VarSep [2]  | 1.43  |
> > | DINo   |  **1.27** |
> >
> > DINo is at the same level of performance and even outperforms the previous state-of-the-art [2] in prediction performance. This demonstrates DINo's potential to handle complex real-world spatiotemporal dynamics. We also provide some visualizations of DINo's [train (*link*)](https://anonymous.4open.science/r/DINo-D752/visu/dino_sst_tr.pdf) and [test (*link*)](https://anonymous.4open.science/r/DINo-D752/visu/dino_sst_ts.pdf) predictions. We make two observations. First, DINo fits very accurately the train data. Second, on the test, we observe that the dynamics on low frequencies seem to be correctly modeled while the prediction of high frequencies dynamics are less accurate.
> > Larger scale experiments would be required to effectively evaluate the model performance on this challenging dataset. Given the complexity of the data, this is out of the scope of the paper.
> > Moreover, there is certainly room for further improvements of the above model, with more time available.
> > Yet, these experiments already demonstrate that DINo behaves competitively w.r.t the previous state-of-the-art.
> >
> > [1] De Bézenac et al. Deep Learning For Physical Processes: Incorporating Prior Scientific Knowledge. ICLR 2018.\
> > [2] Donà et al. PDE-Driven Spatiotemporal Disentanglement. ICLR 2021.\
> > [3] Madec et al. NEMO ocean engine. Technical Report 27, Scientific Notes of Climate Modelling Center, Institut Pierre-Simon Laplace (IPSL). Zenodo.\
> > [4] Yildiz et al. ODE$^2$VAE: Deep generative second order ODEs with Bayesian neural networks. NeurIPS 2019.

---

### Official Review · Reviewer_P8h2 · 2022-10-23

**Confidence:** 4
**Correctness:** 4
**Technical Novelty And Significance:** 4
**Empirical Novelty And Significance:** 3
**Recommendation:** 6

**Clarity, Quality, Novelty And Reproducibility:**

Clarity: excellent

Quality: excellent

Novelty: good

Reproducibility: good, it provides the code. It would be nice to be more clear about the in-t and out-t range.

**Strength And Weaknesses:**

Strength:
The paper is well motivated and clearly written. The method is novel, and the experiment evaluation is thorough. The quality of the paper is very high.

Weaknesses:
1. It is not clear how the method's error accumulate in the long-term evolution for out-t scenario. The long-term evolution is a key challenge in neural-based solvers. A model with excellent short-term error does not necessarily mean a good long-term error (since the error can accumulate and the model can overfit to short-term behavior.

Therefore, it would be nice and strengthen the paper, if the authors show how different models behave where the out-t is range is significantly greater than the in-t (e.g. 5 times, 10 times, 50 times the length). A figure showing the error accumulation is also nice.

2. It is not clear how time consuming is the method. The auto-decoding may need multiple steps to infer the correct alpha. In addition, the latent evolution may be more time consuming that regular time intervals. It would be great if the authors can report the runtime (in seconds) of each method.
2. The author mentioned FNO in the introduction. it would be nice to compare with the proposed method with the strong baseline of FNO.

**Summary Of The Paper:**

This paper proposes DINO, a model that generalizes to arbitrary spatial and temporal resolutions, beyond the spatial and temporal samples in training. This is achieved via a combination of autodecoding, dynamics model, and hypernetwork with amplitude modulation. It shows in experiment that DINO achieves in general superior performance in generalizing to novel time interval and spatial resolutions, compared to strong baselines.

**Summary Of The Review:**

In summary, the paper is well-motivated, well-written, novel, and solid (through thorough experiments). The paper would be strengthened if the weaknesses is addressed.

---

> ### Author Response · Authors · 2022-11-14
> **Response to Reviewer P8h2**
>
> We thank the reviewer for the thoughtful review. We address the concerns below. We will complement it later with a revision of our paper.
>
> ### 1. How Does DINo's Error Accumulate in the Long Term?
>
> Following the reviewer's suggestion, we provide an analysis of error accumulation over time for $T'=T+\Delta T$ where $\Delta T\in\\{5T, 10T, 50T\\}$.  We generate a Navier-Stokes dataset with longer trajectories and report MSE for $\Delta T\in\\{T, 5T, 10T, 50T\\}$ and $T'=T+\Delta T$ in the following table. Note that $\Delta T = T$ is the setting in our initial submission ($T' = 2T$).
>
> | Out-t i.e. $]T, T+\Delta T]$ | Model | $\Delta T=T$ | $\Delta T=5T$ | $\Delta T=10T$ | $\Delta T=50T$ |
> |--------| --------   | -------- | --------   |--------   | --------   |
> | Navier-Stokes, $s = 5$\% | DINo  | **2.017e-3** | **4.895e-3** | **1.209e-2** | **1.440e-1**|
> |                   | I-MP-PDE| 8.387e-3 | 3.580e-2 | 3.356e-1 | 4.031e+1 |
> | Navier-Stokes, $s = 100$\% | DINo  | **4.617e-4** | **2.082e-3** | **6.901e-3** | **1.215e-1** |
> |                   | MP-PDE| 5.251e-4 | 3.524e-2 | 3.339e-1 | 9.755e+1 |
>
> We observe that DINo's MSE in long-term forecasting is more than an order of magnitude smaller than for (I-)MP-PDE. This demonstrates the extrapolation abilties of our model. We will include this new analysis in our revision.
>
> ### 2. Time Efficiency and Runtime of Auto-Decoding and Dynamics Unrolling
>
> * *Number of steps to infer $\alpha_0$.* Given a decoder and an observation frame $v_0$, finding $\alpha_0$ corresponds to solving an inverse problem (Eq (3) in the paper). At inference, we use 300 steps to infer $\alpha_0$; using less steps is possible but results in slight underfitting. This represents 2.76s for 64 trajectories on a single Tesla V100 Nvidia GPU. Note that, as we unroll dynamics in the latent space, there is no need to relearn $\alpha_t$ when $t>0$. Moreover, this differs from training, where $\alpha_t$ is continuously optimized $\forall t\in[0,T]$ within the train horizon, alternatively with our INR decoder. Overall, we trained MP-PDE and DINo for $\sim$7 days such that there is no major additional temporal training cost for DINo.
> * *Using neural ODEs in the latent space.* Unrolling the dynamics with a neural ODE is efficient (0.35s for unrolling 19 time predictions for 64 trajectories on a single Tesla V100 Nvidia GPU). Indeed, the latent space is small (at most 100 dimension) and the dynamics models is a simple 4-layer MLP. With the same latent dynamics model, using an RK4 numerical scheme only incurs four additional function evaluations over a discretized alternative e.g. standard ResNet. This incurs a minor computational cost but enables DINo to operate at different temporal resolutions (Section 5.2, paragraph **Finer time resolution**), unlike e.g. MP-PDE.
>
> In comparison, the official code of MP-PDE takes 312s for inference on the same hardware for the same number of trajectories (vs 3s for DINo). MP-PDE requires building an adjacency matrix and incurs for this reason a high memory cost, especially as the number of nodes increases. Interpolation also significantly increases inference time. This is not the case for DINo, which is faster.
>
> We will incorporate this discussion in the appendix of our revision.
>
> ### 3. FNO Baseline
>
> We actually included a FNO baseline, MNO, which corresponds to the auto-regressive formulation of FNO (a.k.a. FNO-2D) that is designed and required for temporal extrapolation. We introduced MNO in Section 5.1, paragraph **Baselines**.
>
> ### In-t and Out-t Range
>
> We chose for all datasets $T'=2T$. The range of $T$ depends on the nature of the dataset and is specified in Appendix C; however, we always consider 10 consecutive frames for In-t and 10 more for Out-t. We will clarify this in our revision.

---

### Official Review · Reviewer_swAv · 2022-10-25

**Confidence:** 3
**Correctness:** 3
**Technical Novelty And Significance:** 3
**Empirical Novelty And Significance:** 2
**Recommendation:** 6

**Clarity, Quality, Novelty And Reproducibility:**

The paper writing is clear and the quality is good. The dataset is publicly available.  The experiments setup and parameters are provided in the paper and appendix. The data and code is available in an anonymous link.


**Strength And Weaknesses:**

Strength:
The problem is interesting and challenging.  The forecasting model for continuous space-time is important for many real-world tasks.
The model does not rely on fixed discretization on training and test data and can extrapolate at arbitrary spatial and temporal locations.
The extensive experiments on the simulated dataset validate the model’s performance in modeling the evolution of the dynamic system.

Weakness:
The model decoder only uses the current latent dynamic temporal feature (alpha_t) and ignores previous spatial information. On different spatial locations, the decoder prediction is also independent, which ignores the spatial autocorrelation among samples.
Some baseline for spatiotemporal interpolation and extrapolation with common machine learning models is lacking, such as the Gaussian process.
The bi-level optimization training for the dynamic model and auto-decoder seems hard to converge. Some analysis on the training and convergence is needed.
More evaluations on some real-world spatiotemporal datasets like weather forecasting, air quality forecasting would be interesting to show the generalization capability on real dynamic problems.


**Summary Of The Paper:**

This paper proposed a continuous space-time data-driven model for predicting the spatiotemporal evolution of physical phenomena driven by PDE. The method embeds spatial observations independently of their discretization via Implicit Neural Representations and then models continuous-time evolution by a learned latent ODE. The paper claims it can learn from sparse irregular grids or manifolds and extrapolate at arbitrary spatial and temporal locations.

**Summary Of The Review:**

The paper proposes an encoder-decoder framework to address the continuous space-time data-driven model for predicting spatiotemporal evolution, which is important and challenging. The extensive experiments on the simulated dataset validate the model’s performance and the capability to extrapolate at arbitrary spatial and temporal locations. However, there are several concerns to be addressed:

1. The ignorance of spatial autocorrelation in the framework.
2. There is no analysis of the training convergence. How to ensure the encoded feature reflect the spatial
Real-world spatiotemporal forecasting evaluation would be more interesting.

---

> ### Author Response · Authors · 2022-11-14
> **Response to Reviewer swAv - 1/2**
>
> We thank the reviewer for the thoughtful review. We address the concerns below. We will complement it later with a revision of our paper.
>
> ### Taking into Account Previous Spatial Information
>
> > The model decoder only uses the current [$\alpha_t$] and ignores previous spatial information.
>
> This is a standard modeling choice for dynamical systems, more particularly in state-space models [1,2]. $\alpha_t$ is a learned representation of the state of the system; it contains the necessary information to reconstruct the input $v_t$ via decoding. It takes into account previous spatial information by integrating in time the encoding of the initial condition $v_0$, $\alpha_0$, via our ODE-based dynamics model.
>
> We focus on Initial Value Problems (Section 2), where the knowledge of $v_t$ at any time $t$ suffices to infer $v_{t'}$ at any other time $t'>t$. In this case, the model does not need to take several consecutive time observations $(v_{t})_{0 \leq t \leq C}$ as input. Note that the current Markovian setting is a common assumption in the domain [3]. We will update the description of our setting in Section 2 and the model description in Section 4 to clarify this point.
>
> [1] Ljung and Glad. Modeling of Dynamic Systems (Chapter 3.4). Prentice Hall, 1994.\
> [2] Fraccaro. Deep Latent Variable Models for Sequential Data. PhD thesis, Danmarks Tekniske Universitet, 2018.\
> [3] Li et al. Learning Dissipative Dynamics in Chaotic Systems. NeurIPS 2022.
>
> ### Autocorrelation Between Spatial Locations
>
> > On different spatial locations, the decoder prediction is also independent, which ignores the spatial autocorrelation among samples.
>
> If we understood correctly, the reviewer means that, within each frame, predictions across spatial locations are independent from each other. On the contrary, DINo models spatial autocorrelation within each frame via Eq. (4). Our decoder's predictions at all spatial locations $x\in\Omega$ depend on the same latent vector $\alpha_t$: $$\tilde{v}(x) = g_{\phi}(\alpha_{t})(x).$$ Therefore, the decoder's outputs across spatial locations are not independent. DINo operates in the frequency domain via our FourierNet implementation: $\alpha_t$ represents learned amplitudes of a spatial Fourier basis ($s_{\gamma^{(m)}}$ in Eq. (8)), learned over *all* observations. We will clarify our description of the decoder in Section 4.2 to better highlight this fact.
>
> ### Baselines
>
> > Some baseline for spatiotemporal interpolation and extrapolation with common ML models is lacking, such as the Gaussian process.
>
> We already considered three strong ML baselines for interpolating and extrapolating spatiotemporal data: bicubic interpolation combined to a forecaster as standardly done in the domain [4] (here, I-MP-PDE), the neural operator DeepONet, and standard INRs (SIREN). All three are established and state-of-the-art methods in the recent ML and ML $\times$ Physics literature. We looked for recent papers on Gaussian Processes for PDE modeling, but we did not find a method that could be directly applied to our setting. We would be interested in suggestions from the reviewer.
>
> [4] Chae et. al. PM10 and PM2.5 real-time prediction models using an interpolated convolutional neural network. Scientific Reports, 2021.
>
> ### Convergence Analysis
>
> We stress that DINo can be trained in a two-stage process, like [5] does for static data. Training would consist in first applying the auto-decoding scheme to obtain convenient representations $\alpha_t$ for all training frames; then learning to predict the representations from the initial condition $\alpha_0$. For simplicity, we solve the two stages in parallel. This significantly differs from bi-level, min-max training since the objectives are not in conflict. Consequently, we experimentally found neither stability nor convergence issue.
>
> To demonstrate this, we conducted an additional experiment. We track the auto-decoding loss $\ell_{\text{dec}}$ and the dynamics loss $\ell_{\text{dyn}}$ across epochs on Navier-Stokes ($s=100$\%) in this [figure (*link*)](https://anonymous.4open.science/r/DINo-D752/visu/learning_curve.pdf), which we will incorporate in the appendix of our revision. We observe that both losses do not diverge throughout training.
>
> [5] Dupont et al. From data to functa: Your data point is a function and you can treat it like one. ICML 2022.

---

> > ### Author Response · Authors · 2022-11-14
> > **Response to Reviewer swAv - 2/2**
> >
> > ### Real-World Dataset
> >
> > Our PDE datasets all correspond to simulation of complex spatiotemporal systems involved in many real-world applications (e.g. atmosphere modeling or weather forecasting). Simulated data is currently standard for benchmarking data-driven PDE forecasters in the ML community. Our datasets have *similar or higher difficulty* than recent papers at top-tier ML conferences, e.g. [6,7,8,9]. Note that our Shallow-Water spherical PDE is not trivial to model without specialized encoder architectures and is relevant for modeling phenomena on the Earth; to the best of our knowledge, it is modeled *for the first time* with a fully data-driven approach. If time permits, we will attempt to evaluate DINo on data collected in a real-world setting, as recommended by the reviewer.
> >
> > [6] Li et al. Fourier Neural Operator For Parametric Partial Differential Equation. ICLR 2021.\
> > [7] Yin et al. Augmenting Physical Models with Deep Networks for Complex Dynamics Forecasting. ICLR 2021 (oral).\
> > [8] Brandstetter et al. Message Passing Neural PDE Solver. ICLR 2022 (spotlight).\
> > [9] Boussif et al. MAgNet: Mesh Agnostic Neural PDE Solver. NeurIPS 2022.

---

> > > ### Author Response · Authors · 2022-11-18
> > > **Second Response to Reviewer swAV: Additional Real-World Experiment and Model Extension**
> > >
> > > As recommended by the reviewer, we include the following changes in our revision. We would be happy to answer any further question.
> > >
> > > **Real-world data.** We evaluate DINo on real-world data. Following [1,2], we model the Sea Surface Temperature (SST) of the Atlantic ocean, derived from the data-assimilation engine [NEMO (Nucleus for European Modeling of the Ocean) (link)](https://data.marine.copernicus.eu/product/GLOBAL_ANALYSIS_FORECAST_PHY_001_024/description) using E.U. Copernicus Marine Service Information [3]. Accurately modeling SST dynamics is critical in weather forecasting or planning of coastal activities. This problem is particularly challenging as SST dynamics are only partially observed: several unobserved variables affecting the dynamics (e.g. the sea water flow) need to be estimated from data.
> > >
> > > **Incorporating consecutive time steps.** To model SST which includes non-Markovian data, we modify our dynamics model in a similar fashion to [4] to integrate a history of several consecutive observations. This experiment confirms that our space- and time-continuous framework can easily be extended to incorporate other rich temporal models. In more details, we define a neural ODE over an augmented state $[\alpha_t, \alpha_t']$ where $\alpha_t$ is our auto-decoded state and $\alpha'_t$ is an encoding of $\tau=4$ past auto-decoded observations via a NN $c$. We adjust our inference and training settings as in Appendix Section E.3 of the revision.
> > >
> > > **Results.**
> > > For this experiment, we train DINo on trajectories collected from three geographical zones (17 to 20) and fix $\tau=4$ and $T=9$ days. We report MSE for test trajectories within the time interval $[\tau, T]$ for DINo and VarSep [2], the current state-of-the-art on SST, adapted to our inference procedure and retrained on the same training data.
> > >
> > > | Method  | MSE |
> > > |--------| --------   |
> > > | VarSep [2]  | 1.43  |
> > > | DINo   |  **1.27** |
> > >
> > > DINo is at the same level of performance and even outperforms the previous state-of-the-art [2] in prediction performance. This demonstrates DINo's potential to handle complex real-world spatiotemporal dynamics. We also provide some visualizations of DINo's [train (*link*)](https://anonymous.4open.science/r/DINo-D752/visu/dino_sst_tr.pdf) and [test (*link*)](https://anonymous.4open.science/r/DINo-D752/visu/dino_sst_ts.pdf) predictions. We make two observations. First, DINo fits very accurately the train data. Second, on the test, we observe that the dynamics on low frequencies seem to be correctly modeled while the prediction of high frequencies dynamics are less accurate.
> > > Larger scale experiments would be required to effectively evaluate the model performance on this challenging dataset. Given the complexity of the data, this is out of the scope of the paper.
> > > Moreover, there is certainly room for further improvements of the above model, with more time available.
> > > Yet, these experiments already demonstrate that DINo behaves competitively w.r.t the previous state-of-the-art.
> > >
> > > [1] De Bézenac et al. Deep Learning For Physical Processes: Incorporating Prior Scientific Knowledge. ICLR 2018.\
> > > [2] Donà et al. PDE-Driven Spatiotemporal Disentanglement. ICLR 2021.\
> > > [3] Madec et al. NEMO ocean engine. Technical Report 27, Scientific Notes of Climate Modelling Center, Institut Pierre-Simon Laplace (IPSL). Zenodo.\
> > > [4] Yildiz et al. ODE$^2$VAE: Deep generative second order ODEs with Bayesian neural networks. NeurIPS 2019.

---

### Author Response · Authors · 2022-11-14
**First Response to Reviewers**

We thank the reviewers for their thoughtful remarks, constructive feedback and overall positive comments. We are encouraged that they acknowledge the technical novelty and significance of our contributions and the challenge and interest of space- and time- continuous modeling (R.swAv, R.eLsK, R.4pqH). They also found our paper of "high quality", "well-written" (R.P8h2, R.eLsK, R.4pqH), "well-motivated" (R.P8h2), reproducible (all reviewers) and with "strong baselines" (R.P8h2).

We provide a first response to each reviewer individually. We will follow up this initial response with a revision of our paper, taking into account this feedback. We look forward to further discussing with the reviewers.

Our first response contains in particular the following key points.

* **Datasets** (R.swAV, R.eLsK, R.4pqH). Our PDE datasets all correspond to simulation of complex spatiotemporal systems involved in many real-world applications (e.g. atmosphere modeling or weather forecasting). Simulated data is currently standard for benchmarking data-driven PDE forecasters in the ML community. Our datasets have *similar or higher difficulty* than recent papers at top-tier ML conferences, e.g. [1,2,3,4]. Note that our Shallow-Water spherical PDE is not trivial to model without specialized encoder architectures and is relevant for modeling phenomena on the Earth; to the best of our knowledge, it is modeled *for the first time* with a fully data-driven approach. If time permits, we will attempt to evaluate DINo on data collected in a real-world setting, as recommended by R.swAV and R.eLsK.
* **Extensions of DINo** (R.swAv, R.eLsK). Interesting extensions of DINo could indeed be considered, such as integrating several time-points, incorporating stochasticity in the dynamics and modeling the multi-modality in latent vectors. However, this is beyond the scope our paper: we aimed, like most prior work, at modeling deterministic Initial Value Problems (Section 2) to highlight our central contribution -- combining a spatial INR (auto-)decoder with a dynamical model. This contribution does not rely on the nature and complexity of the latent dynamical model which could easily incorporate other state-of-the-art temporal models. Hence, we leave these exciting extensions for future work.
* **Additional experimental studies**. Following the reviewers' suggestions, we conduct a diverse set of new experiments: we analyze the stability of training convergence (R.swAv), study long-term extrapolation and report runtime (R.P8h2) and better show the difficulty of our datasets by computing interpolation errors and highlighting failure cases (R.4pqH).

[1] Li et al. Fourier Neural Operator For Parametric Partial Differential Equation. ICLR 2021.\
[2] Yin et al. Augmenting Physical Models with Deep Networks for Complex Dynamics Forecasting. ICLR 2021 (oral).\
[3] Brandstetter et al. Message Passing Neural PDE Solver. ICLR 2022 (spotlight).\
[4] Boussif et al. MAgNet: Mesh Agnostic Neural PDE Solver. NeurIPS 2022.

---

> ### Author Response · Authors · 2022-11-18
> **Second Response to Reviewers: Revision, Additional Real-World Experiment and Model Extension.**
>
> We complement our first response with the following elements. With these additional studies, we hope to have answered the remaining questions on our submission. If not, we would be glad to answer any further question.
>
> * **Paper revision.** We provide the revision of our paper which takes into account the reviewers' suggestions. Main modifications are colored in *magenta*.
> * **Additional real-world experiment (R.swAV, R.eLsK).** We evaluate DINo on real-world data. Following [1,2], we model the Sea Surface Temperature (SST) of the Atlantic ocean, derived from the data-assimilation engine [NEMO (Nucleus for European Modeling of the Ocean) (link)](https://data.marine.copernicus.eu/product/GLOBAL_ANALYSIS_FORECAST_PHY_001_024/description) using E.U. Copernicus Marine Service Information [3]. Accurately modeling SST dynamics is critical in weather forecasting or planning of coastal activities. This problem is particularly challenging as SST dynamics are only partially observed: several unobserved variables affecting the dynamics (e.g. the sea water flow) need to be estimated from data. In the short amount of time available for this additional experiment, we obtained encouraging results and even outperformed the previous state-of-the-art [2] in prediction performance. In our opinion, this already demonstrates DINo's potential to handle complex real-world spatiotemporal dynamics. We detail our state-of-the-art results for this challenging real-world dataset in the [response to Reviewers swAV and eLsK (link)](https://openreview.net/forum?id=B73niNjbPs&noteId=iiL2Fl_9K3) and in Appendix E.3 in our revision.
> * **Incorporating consecutive time steps (R.swAV, R.eLsK).** To model the non-Markovian data of SST, we slightly modify our dynamics model in a similar fashion to [4] in order to integrate a history of several consecutive observations as conditioning observations. This experiment confirms that our space- and time-continuous framework can easily be extended to incorporate other rich temporal models.  We detail our model extension in the [response to Reviewers swAV and eLsK (link)](https://openreview.net/forum?id=B73niNjbPs&noteId=iiL2Fl_9K3) and in Appendix E.3 in our revision.
>
> [1] De Bézenac et al. Deep Learning For Physical Processes: Incorporating Prior Scientific Knowledge. ICLR 2018.\
> [2] Donà et al. PDE-Driven Spatiotemporal Disentanglement. ICLR 2021.\
> [3] Madec et al. NEMO ocean engine. Technical Report 27, Scientific Notes of Climate Modelling Center, Institut Pierre-Simon Laplace (IPSL). Zenodo.\
> [4] Yildiz et al. ODE$^2$VAE: Deep generative second order ODEs with Bayesian neural networks. NeurIPS 2019.

---

### Author Response · Authors · 2022-12-06
**A Gentle Reminder**

Dear Reviewers swAv, P8h2 and eLsK,

We have carefully addressed all the points raised in your reviews. As requested, we added several new experiments (including one on real-world data) and provided further details and explanations.

We would be grateful if you could acknowledge that our responses have been taken into account. We would be glad to complement them if necessary.

Thank you again for your useful feedback on our work.

---

> ### Comment · Reviewer_eLsK · 2022-12-07
> **Updates**
>
> Thank you for the thorough reply, and the updates on the paper. I took them into account and changed the given score.

---

### Decision · Program_Chairs · 2023-01-20

**Decision:**

Accept: notable-top-25%

**Justification For Why Not Higher Score:**

Although the paper has a clear merit to be published and presented to the community, there are still reasons which stops me from giving an oral designation. The resulting models are still much smaller scale than many PDEs tackled in real systems. Hence, although the work is interesting its scaling might still be an issue. Moreover, there is no discussion on computational efficiency. These make me question general applicability.

**Justification For Why Not Lower Score:**

The paper has clearly merit to be published in the conference. Moreover, it has a significant contribution to the larger community of ML for physics/ML for science. Hence, it has merit to be a spotlight paper and shared with the community as a short presentation.

**Metareview: Summary, Strengths And Weaknesses:**

The paper is proposing a data-driven forecasting for solution for physical models. The main contribution is using implicit models to extend existing (ir)regular grids to continous domains effectively enabling arbitrary and changing sampling grids for both spatial and temporal domain. To handle the complexity hypernetworks using multiplicating fourier blocks is proposed with seperation of variables enabling effective learning. The model is later tested extensively on a large collection of datasets. The paper was reviewed by 4 experts all uniformly agreed on the merits of the paper. Authors addressed the reviews and there was an active discussion. I strongly recommend authors to go over this lively discussion and integrate them to the paper as much as possible before the camera ready deadline.

**Note From Pc:**

if the above contains the word "oral" or "spotlight" please see: "oral" presentation means -> notable-top-5% and "spotlight" means -> notable-top-25%. As stated in our emails, we are disassociating presentation type from AC recommendations